# The splanchnic mesenchyme is the tissue of origin for pancreatic fibroblasts during homeostasis and tumorigenesis

Lu Han [1], Yongxia Wu[2,3], Kun Fang[4,9], Sean Sweeney[1], Ulyss K. Roesner[1], Melodie Parrish[1], Khushbu Patel[1], Tom Walter[1], Julia Piermattei[1], Anthony Trimboli[3,5], Julia Lefler[1], Cynthia D. Timmers[1], Xue-Zhong Yu [2,3], Victor X. Jin [4,9], Michael T. Zimmermann [3,5,6,7], Angela J. Mathison [6,8], Raul Urrutia [3,5,6,8], Michael C. Ostrowski [1] ✉ & Gustavo Leone [3,5] ✉

Pancreatic cancer is characterized by abundant desmoplasia, a dense stroma composed of extra-cellular and cellular components, with cancer associated fibroblasts (CAFs) being the major cellular component. However, the tissue(s) of origin for CAFs remains controversial. Here we determine the tissue origin of pancreatic CAFs through comprehensive lineage tracing studies in mice. We find that the splanchnic mesenchyme, the fetal cell layer surrounding the endoderm from which the pancreatic epithelium originates, gives rise to the majority of resident fibroblasts in the normal pancreas. In a genetic mouse model of pancreatic cancer, resident fibroblasts expand and constitute the bulk of CAFs. Single cell RNA profiling identifies gene expression signatures that are shared among the fetal splanchnic mesenchyme, adult fibroblasts and CAFs, suggesting a persistent transcriptional program underlies splanchnic lineage differentiation. Together, this study defines the phylogeny of the mesenchymal component of the pancreas and provides insights into pancreatic morphogenesis and tumorigenesis.

Coordinated signaling between the epithelium and the mesenchyme is a recurring theme during different stages of normal development and disease[1,2]. The mesenchyme governs the proper formation of the pancreas during fetal development, and fibroblasts regulate the tumor epithelium behavior during pancreatic cancer progression[3]. Pancreatic ductal adenocarcinoma (PDAC) is one of the most lethal types of malignancy and CAFs can modulate

therapeutic responses[4–6]. Intriguingly, CAFs play opposing roles in promoting and inhibiting pancreatic cancer[7–9]. CAF heterogeneity, with different subtypes displaying distinct gene expression profiles and phenotypes[10], may explain their antonymous impact on PDAC progression. Different origins of tumor cells result in different malignant epithelial phenotypes[11–13], and presumably, different tissue origins of CAFs may similarly lead to CAF heterogeneity and

[1]Department of Biochemistry and Molecular Biology, Hollings Cancer Center, Medical University of South Carolina, 171 Ashley Ave, Charleston, SC 29425, USA. [2]Department of Microbiology and Immunology, Hollings Cancer Center, Medical University of South Carolina, 171 Ashley Ave, Charleston, SC 29425, USA. [3]Medical College of Wisconsin Cancer Center, Medical College of Wisconsin, 8701 Watertown Plank Road, Milwaukee, WI 53226, USA. [4]Division of Biostatistics, Medical College of Wisconsin, 8701 Watertown Plank Road, Milwaukee, WI 53226, USA. [5]Department of Biochemistry, Medical College of Wisconsin, 8701 Watertown Plank Road, Milwaukee, WI 53226, USA. [6]Linda T. and John A. Mellowes Center for Genomic Sciences and Precision Medicine, Medical College of Wisconsin, 8701 Watertown Plank Road, Milwaukee, WI 53226, USA. [7]Clinical and Translational Sciences Institute, Medical College of Wisconsin, 8701 Watertown Plank Road, Milwaukee, WI 53226, USA. [8]Department of Surgery, Medical College of Wisconsin, 8701 Watertown Plank Road, Milwaukee, WI 53226, USA. [9]Present address: Medical College of Wisconsin Cancer Center, Medical College of Wisconsin, 8701 Watertown Plank Road, Milwaukee, WI 53226, USA. ✉e-mail: ostrowsk@musc.edu; gleone@mcw.edu

impact CAF behavior. Potential CAF sources include the pancreatic epithelium (through epithelium-to-mesenchyme transition), bone marrow (through circulation), and pancreatic resident fibroblasts (through proliferation).

In this study, we perform lineage tracing experiments by incorporating various genetic reporters to a genetically engineered mouse model of PDAC previously constructed by our group, the *KPF* model (*Kras*$^{G12D/+}$;*Trp53*$^{frt/+}$;*Pdx1*$^{Flpo/+}$ and Fig. 1a)[14]. In this model, the DNA recombinase *FlpO/Frt* system directs the expression of oncogene *Kras*

and loss of tumor suppressor *Trp53* in the pancreatic epithelium. The *FlpO* allele is inserted into the *Pdx1* locus and its expression is under the control of the endogenous *Pdx1* promoter and enhancer regions, as opposed to a previous transgenic approach[15].

## Results

### Epithelium and bone marrow are minor sources

The trans-differentiation of epithelial cells to a mesenchymal state, typically called epithelium-to-mesenchyme-transition (EMT), is an

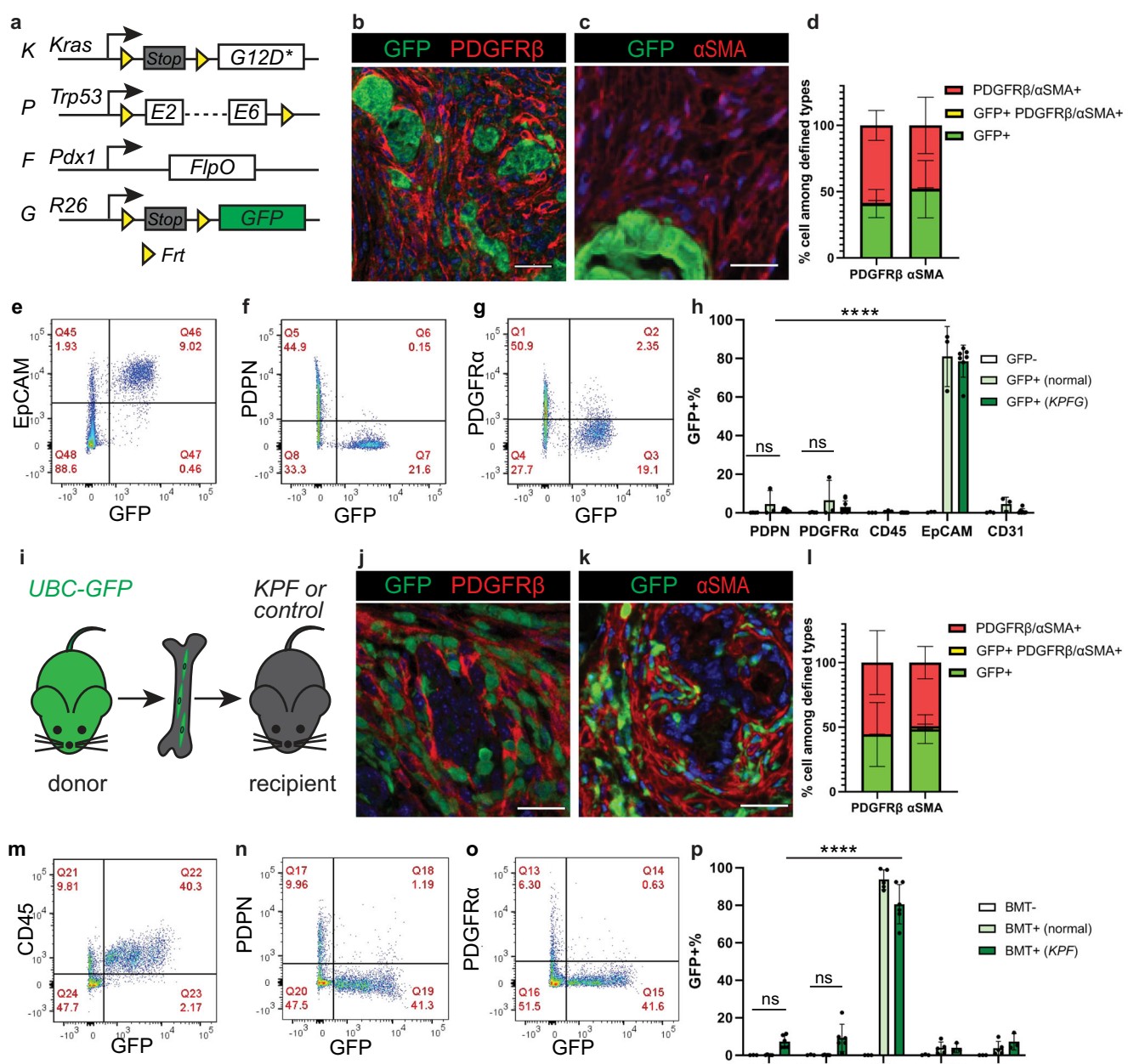

**Fig. 1 | Epithelium and bone marrow minimally contribute to pancreatic cancer associated fibroblasts. a** Schematic of the *KPFG* mouse model. **b–d** Co-immunostaining in the *KPFG* pancreas and quantification of cells that are either single positive or double positive for defined markers. *n* = 7 mice for PDGFRβ analysis; *n* = 8 mice for αSMA analysis. **e–h** Flow cytometry analysis of dissociated *KPFG* pancreas and quantification of GFP + cells within populations defined by markers. *n* = 4 mice for GFP-; *n* = 3 mice for GFP + (normal); *n* = 9 mice for GFP + (*KPFG*). ****$p$ < 0.0001; ns for PDPN: $p$ = 0.996; ns for PDGFRα: $p$ = 0.923. **i** Schematic of the bone marrow transplant (BMT) experiment. **j–l** Co-immunostaining in KPF pancreas

transplanted with GFP + bone marrow and quantification of cells that are either single positive or double positive for defined markers. *n* = 5 mice for PDGFRβ analysis; *n* = 3 mice for αSMA analysis. **m–p** Flow cytometry analysis of pancreas from *KPF* mice transplanted with GFP + bone marrow and quantification of GFP + cells within populations defined by markers. *n* = 4 non-transplanted mice; *n* = 5 transplanted normal mice; *n* = 6 transplanted *KPF* mice. Data are mean ± SD. ****$p$ < 0.0001; ns for PDPN: $p$ = 0.744; ns for PDGFRα: $p$ = 0.340. **h, p** Two-way ANOVA test with Tukey's multiple comparisons was performed. ns, not significant. Scale bars: 30 μm. Source data are provided as a Source data file.

essential embryonic developmental process, also implicated in cancer progression[16]. Previous work showed that some pancreatic tumor cells express fibroblast markers[17]. Therefore, we first sought to assess whether EMT contributes to CAFs in PDAC. To this end, we performed cell tracing experiments incorporating a FlpO-dependent $R26^{GFP}$ reporter[18] into the KPF model (referred to as the KPFG model) (Fig. 1a). This took advantage of the epithelium specific expression of the FlpO to recombine the $R26^{GFP}$ reporter allele, resulting in permanent GFP expression in the epithelium and its descendants. In this current study, fibroblasts in the adult pancreas are defined as cells located in the stroma or interstitium, positive for markers including VIMENTIN (VIM), PDGFRα, PDGFRβ, PDPN or αSMA, and negative for other lineage markers. Despite robust GFP labeling of the epithelium, GFP + cells were only rarely observed in the tumor stroma. This observation is different from a previous report[17], potentially due to differences in mouse model systems. Consistent with this observation, immunohistochemistry (IHC) staining showed few cells co-expressing GFP and fibroblast markers PDGFRβ and αSMA (Fig. 1b–d, Supplementary Fig. 1a, b). We also dissociated pancreata into single cells and performed flow cytometry analysis with cell-type-specific markers (Fig. 1e–h, Supplementary Figs. 1e–k, 8a). This analysis showed that while more than 80% of EpCAM+ epithelial cells were labeled with GFP, very few PDPN+ or PPDGFRα + CAFs (<5%) were labeled with GFP. Expectedly, hematopoietic cells (CD45+) and endothelial cells (CD31 +) were also rarely labeled with GFP. Similar IHC and flow cytometry analyses performed in normal pancreata only containing the FlpO and the $R26^{GFP}$ reporter alleles showed similar results (Fig. 1h, Supplementary Fig. 1c, d). Together, these data suggest that epithelium only rarely gives rise to resident fibroblasts and CAFs in normal and tumor-bearing mice, respectively.

Bone marrow-derived cells have been suggested to migrate, populate and give rise to CAFs in a number of cancer types, including pancreatic cancer[19]. Thus we considered whether bone marrow cells are a significant source of CAFs in our KPF model. To this end, bone marrow cells from UBC-GFP + donor mice with ubiquitous expression of GFP[20] were transplanted into irradiated KPF mice (Fig. 1i). This transplantation method is well-established for replenishing the bone marrow[21–24]. The long-term mouse survival (data now shown) and persistence of GFP + blood cells (Supplementary Fig. 1u) confirmed successful bone marrow engraftment in our current study. Pancreata were then harvested from these chimeric mice with established PDAC. GFP-labeled cells were more abundant in the KPF pancreata than in normal pancreata from littermates (Supplementary Fig. 1m, n). Despite such abundance, only a small number of cells co-expressed GFP and fibroblast markers αSMA and PDGFRβ (Fig. 1j–l, Supplementary Fig. 1o, q–r). Further, IHC and flow cytometry analysis with additional cell markers revealed that the vast majority of GFP + cells were hematopoietic cells (Fig. 1m–p, and Supplementary Figs. 1l, p, s–t, 8b). From these data, we conclude that bone marrow is also a minor (less than 10%) tissue source of resident fibroblasts and CAFs.

## ISL1 + mesenchyme gives rise to TRFs
Fibroblasts in the normal pancreas can be non-lipid storing[25] or lipid storing, the latter often referred to as stellate cells[26]. In this current study, all fibroblasts in the normal pancreas will be referred to as tissue-resident fibroblasts (TRFs). TRFs have been suggested to give rise to CAFs based on computational modeling[27] and in vitro cultures[28]. Recently, lineage tracing studies showed that a subset of TRFs contribute to about half of the CAF pool[25] while stellate cells contribute to a minor subset of pancreatic CAFs[29]. We sought to first identify the fetal progenitors of TRFs. Fibroblasts are generally considered to be derived from the mesoderm during fetal development. After gastrulation, the mesoderm undergoes a series of diversification into different lineages including the splanchnic mesenchyme[30]. The splanchnic mesenchyme is a layer of mesoderm adjacent to the foregut endoderm during fetal development and has been previously shown to give rise to the stromal cells of foregut-derived organs[31,32]. As the pancreas is also a foregut-derived organ, we hypothesized that the splanchnic mesenchyme is the fetal origin of TRFs in the adult pancreas. By analyzing a previously published single-cell RNA sequencing dataset of the developing foregut[33], we noted that the expression of a transcription factor Isl Lim homeodomain 1 (ISL1), along with other established splanchnic markers, was enriched in the splanchnic mesenchyme cluster (Fig. 2a, b, Supplementary Fig. 2a, b). This is consistent with previous studies demonstrating that ISL1 is one of the markers of the splanchnic mesenchyme[33–37]. We confirmed by immunostaining that ISL1 protein was expressed in the mesenchyme surrounding the pancreatic endoderm at mouse embryonic day (E) 9.5 (Fig. 2c, d), consistent with published results[35]. Encouraged by previous studies using $Isl1^{cre}$ to lineage trace the stromal components of foregut-derived organs (lung and esophagus)[36,37], we hypothesized that $Isl1^{cre}$ may be used to target the splanchnic mesenchyme surrounding the pancreas. To test this, we constructed an $I^{re}T$ reporter mouse model, combining the $Isl1^{cre}$ allele[38] and a cre-dependent $R26^{Tomato}$ allele (Fig. 2g). This genetic approach allows recombination of the $R26^{Tomato}$ allele in ISL1 expressing cells, thus irreversibly labeling all their descendants with Tomato expression. $I^{re}T$ alleles labeled the splanchnic mesenchyme surrounding the pancreatic epithelium at E12.5 (Fig. 2e, f, Supplementary Fig. 2c–e). Lineage tracing of these Tomato+ cells showed persistence of labeled cells in the adult pancreas. Tomato+ cells were interspersed in the parenchyma of the pancreas and in perivascular regions. They lacked expression of epithelial marker ECAD and instead expressed fibroblast markers VIM and PDGFRβ, suggesting their fibroblast identity (Fig. 2h–k, Supplementary Fig. 2f). Flow cytometry analysis of dissociated adult pancreas showed about 80% overlap of Tomato and two additional fibroblast markers PDPN and PDGFRα (Fig. 2l–n, Supplementary Fig. 8c). Such overlap was significantly higher than in any other major cell types in the pancreas (Fig. 2n). This is consistent with immunostaining on tissue sections showing minimal overlap between Tomato and a macrophage marker F4/80 (Supplementary Fig. 3a–c). In addition, we cultured fibroblasts from these pancreata in vitro and the vast majority of fibroblasts expressed Tomato (Supplementary Fig. 2g–j). Together, these data indicate that ISL1 expressing splanchnic mesenchyme is the fetal origin of TRFs in the normal adult pancreas.

## ISL1 + mesenchyme gives rise to CAFs
We next investigated the possibility that ISL1 + splanchnic mesenchyme may also give rise to pancreatic CAFs. To test this hypothesis, we combined the $I^{re}T$ reporter alleles with the KPF alleles (Fig. 3a). This approach capitalized on the independent functions of FlpO/Frt and Cre/LoxP as independent DNA recombinase systems, allowing us to lineage trace Tomato labeled mesenchymal cells (via $I^{re}T$) during tumorigenesis (via KPF). We refer to this dual DNA recombinase model as "KPFI$^{cre}$T". While 30-day-old KPFI$^{re}$T mice mainly exhibited pancreatic intra-epithelial neoplastic (PanIN) lesions, 50-day-old mice exhibited PDAC. We examined the epifluorescence of Tomato in dissected pancreata. In the main body of the pancreas, we observed minimal signal in the normal pancreas, local puncta of signal at PanIN stage, and a marked increase of signal throughout the entire pancreas at PDAC stage (Fig. 3b, c). Sectioning and staining of these pancreata confirmed local expansion of Tomato+ cells circumscribing early PanIN lesions, and pervasive expansion of Tomato+ cells in PDAC (Fig. 3d, e, Supplementary Fig. 3g, h), suggesting a gradual expansion of Tomato labeled cells accompanying tumor progression. Co-staining of known markers was performed to rigorously examine cell types expressing the reporter. This showed that Tomato+ cells expressed fibroblast markers VIM, PDGFRβ, and αSMA, and constitute the majority of CAFs (Fig. 3f–i). Whole mount staining of thin slices

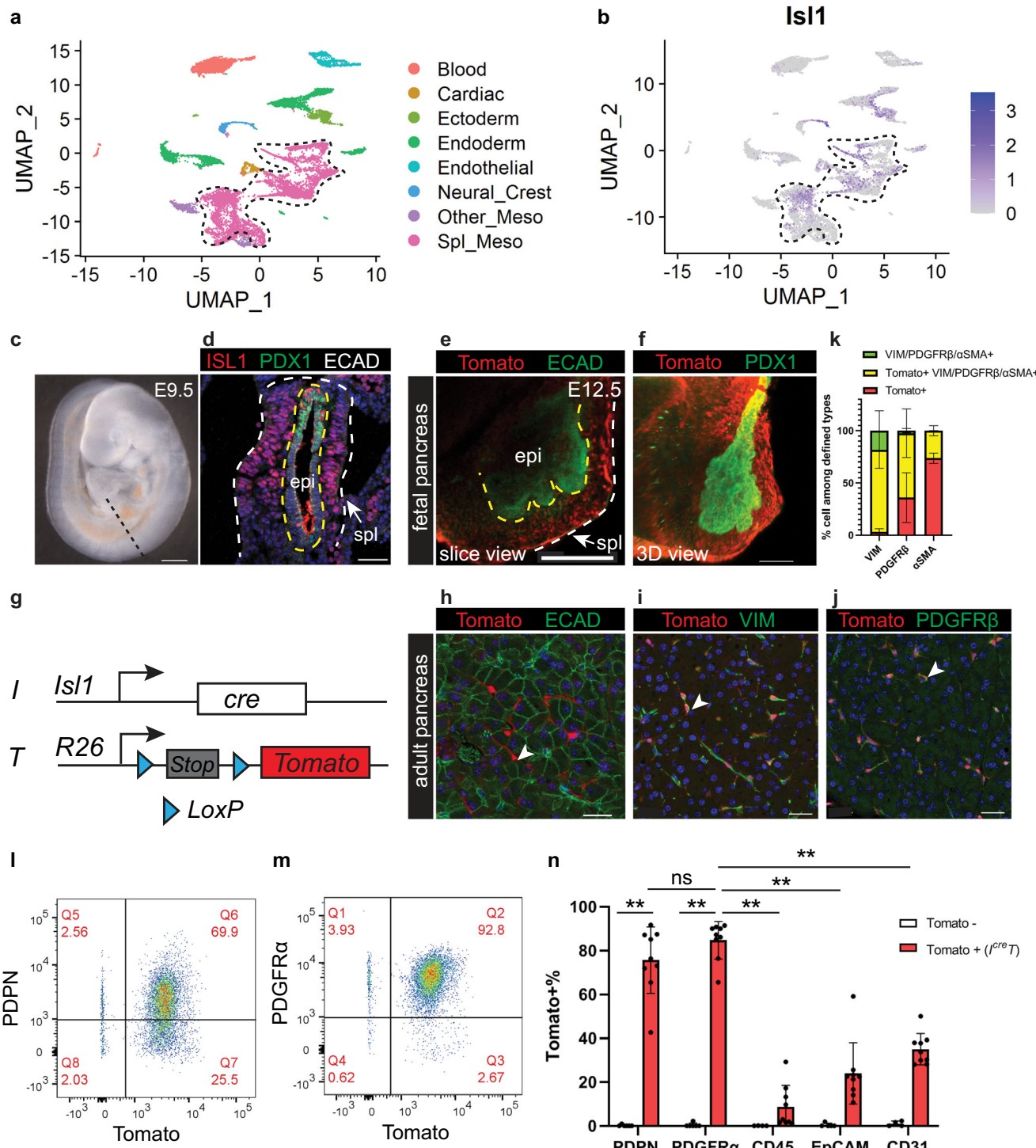

**Fig. 2 | Tissue-resident fibroblasts in the adult pancreas are derived from ISL1 expressing splanchnic mesenchyme during fetal development. a** Clustering analysis of single-cell transcriptome profiles of E9.5 mouse foregut cells. **b** *Isl1* expression projected on clusters in (**a**). Dashed lines delineate the splanchnic mesenchyme cluster. **c** Lateral view of an E9.5 mouse embryo. Dashed line indicates a transverse section at the pancreatic level. **d** Immunostaining on mouse foregut indicated in (**c**). *n* = 3 embryos. **e, f** Whole mount immunostaining of a dissected pancreas. Yellow dashed lines delineate the epithelium, while the yellow and the white lines delineate the splanchnic mesenchyme. *n* = 3 pancreata. **g** Schematic of

the I^creT mouse model. **h**–**k** Co-immunostaining of the adult pancreas and quantification of cells that are either single positive or double positive for defined markers. *n* = 5 mice. **l**–**n** Flow cytometry analysis of I^creT adult pancreas and quantification of Tomato+ cells within populations defined by markers. Tomato-, *n* = 5 mice; I^creT, *n* = 9 mice. Data are mean ± SD. Two-way ANOVA test with Tukey's multiple comparisons was performed. ** indicates *p* < 0.0001. ns, not significant, *p* = 0.547. E, embryonic day; epi, epithelium; meso, mesoderm; spl, splanchnic mesenchyme. Arrowheads indicate Tomato+ cells. Scale bars in (**c**–**f**): 100 µm; in (**h**–**j**): 30 µm. Source data are provided as a Source data file.

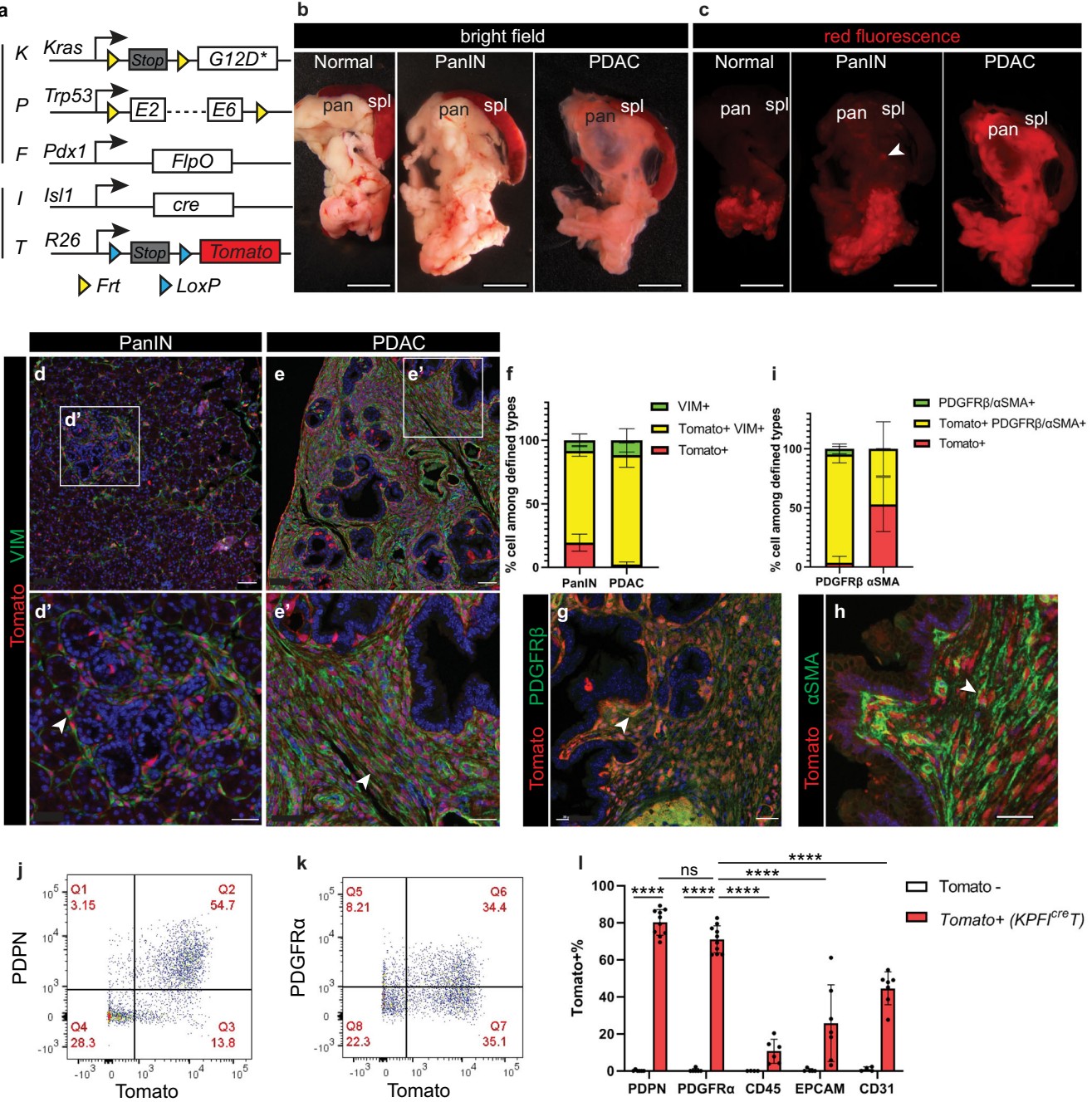

**Fig. 3 | The majority of pancreatic cancer-associated fibroblasts are targeted in the KPFI<sup>cre</sup>T model. a** Schematic of the *KPFF<sup>re</sup>T* mouse model. **b**, **c** Bright field and red epifluorescence views of dissected pancreata and spleens. Normal, $n = 3$ mice; PanIN, $n = 6$ mice; PDAC, $n = 3$ mice. Arrowhead denotes a puncta of red fluorescence. **d–f** Co-immunostaining on pancreatic tissues and quantification of cells that are either single positive or double positive for defined markers. PanIN, $n = 4$ mice; PDAC, $n = 5$ mice. **g–l** Co-immunostaining on pancreatic tissues and quantification of cells that are either single positive or double positive for defined markers.

PDGFRβ, $n = 5$ mice; αSMA, $n = 4$ mice. **j–l** Flow cytometry analysis of a *KPFI<sup>cre</sup>T* pancreas and quantification of Tomato+ cells within populations defined by markers. Tomato-, $n = 5$ mice; *KPFI<sup>re</sup>T*, $n = 7$ mice. Data are mean ± SD. Two-way ANOVA test with Tukey's multiple comparisons was performed. **** indicates $p < 0.0001$; ns, not significant, $p = 0.379$. PanIN, pancreatic intra-epithelial neoplasia; PDAC, pancreatic ductal adenocarcinoma; spl, spleen; pan, pancreas. Arrowheads denote Tomato+ cells. Scale bars in (**b**, **c**): 1000 μm; in (**d**, **e**): 50 μm; in d', e', **g**, **h**: 30 μm. Source data are provided as a Source data file.

across tumor tissues and flow cytometry analysis of dissociated pancreata both showed similar results (Fig. 3j–l, Supplementary Fig. 3d–f, Supplementary Fig. 8c,e). Thus, this data suggest that CAFs are derived from the fetal splanchnic mesenchyme.

**Pulsed labeling of the ISL1 + mesenchyme**
The analysis described above also revealed regionalized expression of Tomato in epithelial cells within the head of the pancreas, regardless of

cancer stage (Fig. 4a, b). This suggested that ISL1 may be dynamically expressed in both mesenchymal and epithelial precursors during fetal development. To test this hypothesis, we utilized an inducible *Isl1<sup>creER</sup>* allele[39] and the *R26<sup>Tomato</sup>* reporter to temporally restrict cre activity. Using this *I<sup>reER</sup>T* model, we activated creER by providing pregnant mice with a single pulse of a tamoxifen (Tam) gavage at E8.5, E9.5, or E10.5, the period when the pancreatic epithelium is specified. We found that activation of *I<sup>reER</sup>T* by Tam at all three time points led to targeting of

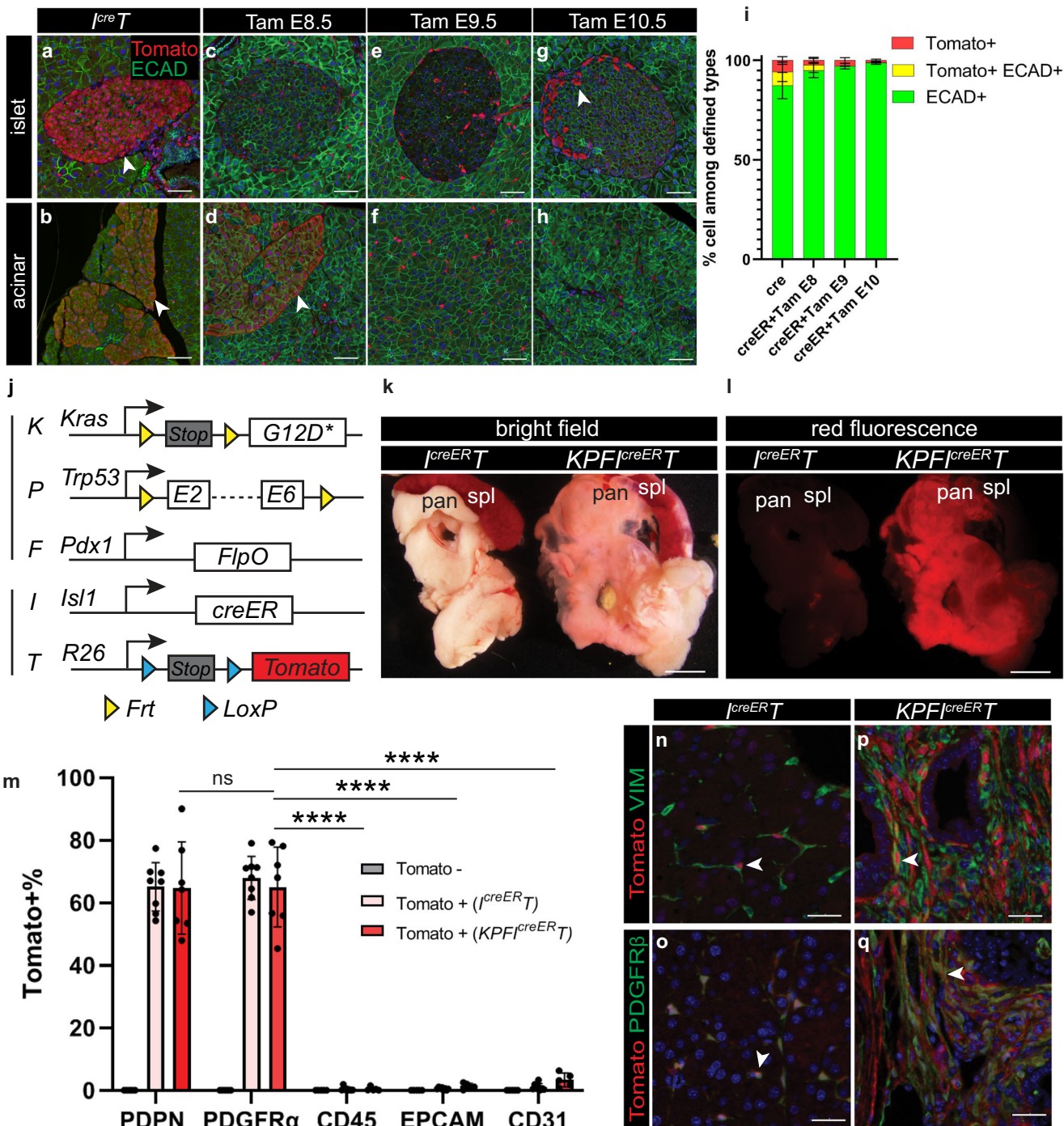

**Fig. 4 | KPFI^creER^T model specifically targets fibroblasts in the pancreas. a–h** Co-immunostaining of pancreatic tissues (focusing on either the islet region or the acinar region) comparing the I^cre^T model and the I^creER^T model treated with Tamoxifen at indicated times points. Arrowheads denote Tomato+ epithelial cells. **i** Quantification of cells that are either single positive or double positive for defined markers in images represented in (**a–h**). I^cre^T, *n* = 5 mice; I^creER^T + Tam E8, *n* = 5 mice; I^creER^T + Tam E9, *n* = 6 mice; I^creER^T + Tam E10, *n* = 4 mice. **j** Schematic the KPFI^creER^T mouse model. **k, l** Bright field and red epifluorescence views of dissected pancreata and spleens. I^creER^T, *n* = 6 mice; KPFI^creER^T, *n* = 7 mice. **m** Quantification of Tomato+ cells within PDPN+, PDGFRα+, CD45+, EpCAM+, or CD31+ cells identified by flow cytometry. Tomato-, *n* = 5 mice; I^creER^T, *n* = 8 mice; KPFI^creER^T, *n* = 7 mice. **n–q** Co-immunostaining of Tomato, VIM, and PDGFRβ on normal and PDAC pancreatic tissues. Arrowheads indicate double positive cells. Data are mean ± SD. Two-way ANOVA test with Tukey's multiple comparisons was performed. **** indicates *p* < 0.0001; ns, not significant, *p* > 0.9999. Spl, spleen; pan, pancreas. Scale bars in **a–h** and **n–q**: 30 μm. Scale bars in **k, l**: 1000 μm. Source data are provided as a Source data file.

pancreatic fibroblasts (Fig. 4c–i, Supplementary Fig. 4a–c). Interestingly, some epithelial cells in the pancreas head were targeted with a E8.5 Tam; essentially no epithelial cells were targeted with a E9.5 Tam; and a small number of the islet endocrine cells were targeted with a E10.5 Tam (Fig. 4c–i). Together, these experiments suggest that a Tam pulse at E9.5 targets the splanchnic mesenchyme and exclusively labels fibroblasts with Tomato expression in the normal adult pancreas.

Having defined the optimal developmental window when pancreatic fibroblasts can be labeled by creER activation, we then used this approach to examine whether the splanchnic mesenchyme

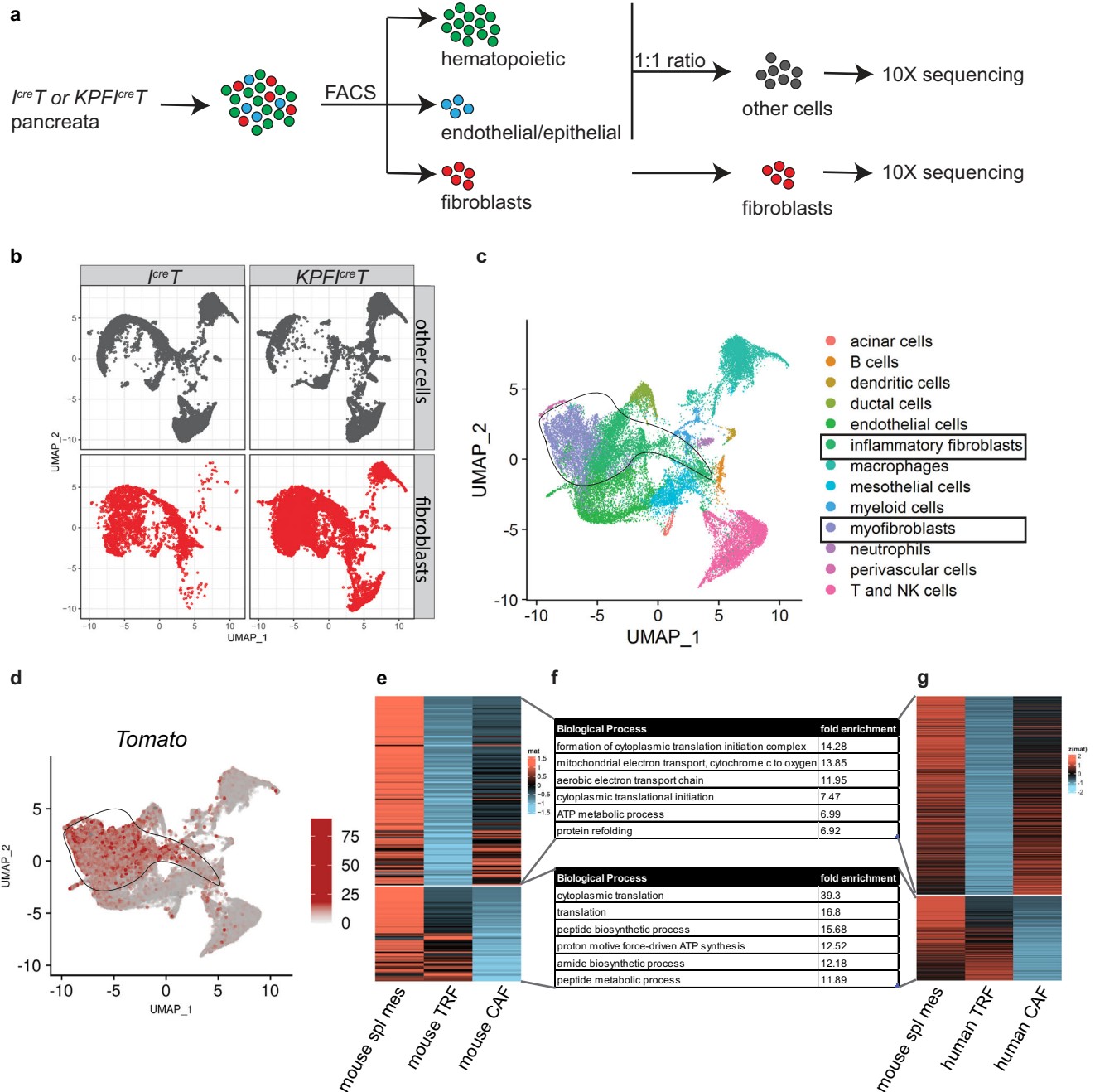

**Fig. 5 | Single-cell analysis of the *KPFI^creT* model suggests a conserved gene signature in the splanchnic lineage. a** Schematic of the single-cell RNA sequencing experimental design. Pancreata were dissociated into single cells and sorted into hematopoietic cells (CD45+), endothelial (CD31+)/epithelial (EPCAM+) cells or fibroblasts (PDPN+ or PDGFRα+, or Tomato+). Hematopoietic cells were combined with endothelial/epithelial cells with 1:1 ratio. Cells were then processed for 10X RNA sequencing. **b** The split view of each sample type in a merged UMAP space. **c** UMAP of cell clusters annotated based on marker gene expression. **d** Tomato gene expression projected onto the cell clusters. **c**, **d** The circled lines delineate the fibroblast cluster. **e** Heatmap of genes highly expressed in both the splanchnic

mesenchyme and CAFs compared to TRFs (upper block), and genes highly expressed in both the splanchnic mesenchyme and TRFs compared to CAFs (lower block). **f** Top biological processes associated with genes identified in mouse analysis (**e**) and human analysis (**g**). **g** Heatmap of genes highly expressed in both the splanchnic mesenchyme and human pancreatic CAFs compared to human pancreatic TRFs (upper block), and genes highly expressed in both the splanchnic mesenchyme and human pancreatic TRFs compared to human pancreatic CAFs (lower block). Spl splanchnic, Meso mesoderm, CAFs cancer associated fibroblasts, TRFs tissue resident fibroblasts.

gives rise to CAFs in *KPF* mice. To this end, pregnant females were pulsed with a single Tam at E9.5 and descendants of splanchnic mesenchyme were traced in *KPFI^creER T* offspring (Fig. 4j). Pancreata were analyzed as before. Similarly, there was a major expansion of the Tomato+ fibroblasts in tumor-bearing mice when compared to littermates with normal pancreata (Fig. 4k, l). Importantly, Tomato expression was highly robust and specific to the fibroblasts, in

comparison to other cell types (Fig. 4m–q, Supplementary Figs. 4d–f, 5a–f, 8c). Tomato labeling was slightly less robust in the tip of the pancreas head than the main body. Expansion of Tomato+ cells was not observed in the lung and the liver (Supplementary Fig. 5j, k). Consistent with their observed expansion in pancreatic tumors, these Tomato+ fibroblasts were proliferative as evidenced by Ki67 expression (Supplementary Fig. 5g–i). From these data, we

conclude that the splanchnic mesenchyme is a major source of fibroblasts in normal and tumor-bearing pancreata.

## Single-cell transcriptome analysis

To examine the expression of Tomato in different types of cells in an unbiased fashion, we performed single-cell RNA sequencing of $F^{re}T$ (normal) and $KPFF^{re}T$ (tumor) pancreata. Different cell populations in the pancreata were enriched by flow sorting based on cell surface marker expressions (Fig. 5a, Supplementary Fig. 8d). Single cells were isolated and their transcriptomes were sequenced. Single-cell expression data were merged into a single-dimension reduction plot (Fig. 5b). We identified all the major cell types in the pancreas based on marker gene expression and cross examination with a published dataset[10] (Fig. 5c, Supplementary Figs. 6, 7a–d). Consistent with previous analysis, Tomato expression was enriched in the fibroblast cluster in comparison to other cell clusters (Fig. 5d). Importantly, both inflammatory CAFs (iCAFs) and myofibroblasts (myCAFs)[10, 28] had Tomato expression (Fig. 5c, d, Supplementary Fig. 7f–i), suggesting both subtypes are derived from the splanchnic mesenchyme, a finding of significant relevance to pancreatic diseases to which these cell types contribute. We also noted that a proportion of iCAFs displayed lower Tomato signal (Fig. 5c, d, Supplementary Data 1). We then compared the gene expression profiles of fibroblasts from the normal pancreata (TRFs), fibroblasts from the PDAC pancreata (CAFs) and the splanchnic mesenchyme from the E9.5 foregut. This analysis identified a group of genes highly expressed in the splanchnic mesenchyme and TRFs, and another group of genes highly expressed in the splanchnic mesenchyme and CAFs (Fig. 5e, Supplementary Data 2, 3, Fig. 7e). A similar analysis performed with human pancreatic TRFs[40] and pancreatic CAFs[41] showed similar patterns (Fig. 5f, g, Supplementary Data 4). The group of genes shared between splanchnic mesenchyme and TRFs are potentially involved in morphogenesis and homeostasis, and the group of genes shared between splanchnic mesenchyme and CAFs may represent fetal programs reactivated during tumorigenesis. We also found fetal splanchnic gene signatures that are present in mouse and human adult pancreatic TRFs and CAFs (Supplementary Fig. 7j, k). This latter data suggest that splanchnic gene signature is persistently expressed through development, into adulthood and cancer in both mouse and human. The expression of these genes may be required to maintain the differentiation and function of these cell populations.

## Discussion

Here, we show that CAFs in pancreatic cancer arise from the fetal splanchnic mesenchyme, with minimal contribution from bone marrow and tumor cells. There is now ample evidence that multiple stromal fibroblast subtypes exist within the pancreatic cancer microenvironment[7], but the source of this heterogeneity was not clear. From cell lineage tracing experiments, we conclude that CAF heterogeneity is unlikely to arise from different tissue origins. Rather, we propose that this heterogeneity stems from differentiation of a single source of the splanchnic mesenchyme that, under the influence of complex microenvironment cues, gives rise to multiple fibroblast subtypes. Our study also defines a clear trajectory of the mesenchymal lineage, revealing intricate tissue dynamics between the epithelial and the mesenchymal components in the pancreas. Early in fetal development, the splanchnic mesenchyme forms a thick layer surrounding the pancreatic epithelium, providing critical early cues for organ development[1]. The splanchnic mesenchyme is subsequently reduced to sparsely distributed resident fibroblasts in the adult pancreas. During tumorigenesis, splanchnic-derived cells dramatically expand to form CAF subtypes that comprise a substantial fraction of the pancreatic cancer tissue mass. Single-cell RNA sequencing suggests that fetal splanchnic gene signatures persist in adult pancreatic fibroblasts. These progenitor expression signatures include expression of stem cell-like genes, reminiscent of cancer stem cell signatures thought to underlie tumor cell survival and drug resistance, and to contribute to tumor cell heterogeneity[11–13]. The functional significance of the expression of these genes in CAFs remains to be determined. While the identification of the tumor cell of origin has led to the design of cell-type-specific activation of oncogenes and development of relevant preclinical models of PDAC[42], similar genetic models that enable a robust and specific targeting of CAFs were previously lacking. By simultaneously targeting both the epithelium and fibroblasts of the pancreas, the *KPFIT* model described here promises to provide important insights into the function and ultimate fates of fibroblasts and tumor epithelial cells during pancreatic cancer progression.

## Methods

### Mice

All studies were conducted under the approval of the Medical University of South Carolina Institutional Animal Care and Use Committee (protocol number: IACUC-2020-00969) or the Medical College of Wisconsin Institutional Animal Care and Use Committee (protocol number: AUA00007337). The *KPF* model (*Pdx1Flpo/+*; *KrasG12D/+*;*p53frt/+*)[14] and *R26GFP* [18] were previously published. *UBC-GFP* (stock no. 004353), *R26Tomato* (stock no. 007914), *Isl1cre* (stock no. 024242), and *Isl1creER* (stock no. 029566) were purchased from the Jackson Laboratory. *UBC-GFP* line was maintained in C57Bl6 background, other lines were maintained in a mixed genetic background including C57BL6, FVB, and Black Swiss. Mice were housed under the following conditions: temperature: 68–74 °F; humidity: 30–60%; light cycle: 12-h light cycle (6a-6p) at the Medical University of South Carolina and 14/10 light cycle (5a-7p and 4a-6p during daylight saving) at the Medical College of Wisconsin. Both males and females of approximately equal ratios were included in the analysis. Veterinary staff examined the mice daily and identified them as "end point" using criteria including ruffled fur, hunched posture, dehydration, abdominal distension, lethargy, a poor body condition score, or when a single tumor reaches 1.6 cm in one direction or cumulative of 1.6 cm in multiple tumors. All mice were euthanized at either end point or earlier as noted in the experiments. Mice were euthanized using either $CO_2$ air displacement followed with cervical dislocation, or cervical dislocation without prior anesthesia for cesarean section.

### Timed mating and tamoxifen administration

For timed mating, one male and one/two female mice were housed in the same cage and plugs were checked every day. The day when a plug was observed was considered embryonic day 0.5. Tamoxifen was dissolved in corn oil and given to plugged females at 0.12 mg/g body weight at indicated time points through oral gavage. To deliver pups that had experienced tamoxifen during gestation, cesarean section was performed. Pups were then fostered with another female that had delivered within 4 days.

### Bone marrow transplantation

Bone marrow recipient mice were maintained as a separate cohort from the rest of the study. *KPF* mice and their littermates were used as recipients. At around 50 days old, recipient mice received 900 cGy of gamma irradiation to eliminate the endogenous bone marrow. On the following day, bone marrow was harvested from arm and leg bones of adult *UBC-GFP* mice after cervical dislocation. Around 600,000 fresh bone marrow cells were transplanted into each recipient mouse through tail vein injection. The recipient mice with *KPF* alleles reached end point around 30 days after the transplantation. Littermates were harvested around the same period as the *KPF* recipients.

## Fresh tissue imaging

After dissection, tissues were imaged under a Nikon dissection microscope using NIS-Elements software. Either bright field or epi-fluorescence of either red or green channels were used for imaging.

## Tissue processing for paraffin embedding

After harvesting, tissues were immediately placed in 10% neutral buffered formalin on a shaker for 24 h at room temperature. Subsequently, tissues were rinsed and transferred to 70% ethanol and given to the Histology Core. Tissues were then verified to be within 4 mm size to allow proper infiltration of processing fluids. Tissues were placed on the Automated Tissue Processor (RTPH-360, General Data), where they underwent immersion of various chemicals, dilutions of ethanol, xylene, and paraffin to have a final product of tissue infiltrated with paraffin. The tissues were then embedded in block form using the embedding center (TEC-II, General Data). The blocks were then cut to 4-μm sections using a microtome (SHURcut A, General Data).

## Immunohistochemistry staining and imaging

Hematoxylin and eosin (H&E) slides were assessed for tumor type/differentiation. Unstained slides were deparaffinized and stained using the Roche Ventana Discovery Ultra Automated Research Stainer (Roche Diagnostics). Heat-induced epitope retrieval (HIER) was performed in EDTA buffer pH 9 (Cat.#S2367, Agilent/Dako) for 32 min at 95 °C and endogenous peroxidase was blocked with a hydrogen peroxide solution after incubation of the first primary antibody. Multiplex immunofluorescence was performed using the OPAL™ multiplexing method based on Tyramide Signal Amplification (TSA). After incubation with primary (Supplementary Data 5) and secondary antibodies, fluorescence signals were generated using the following Akoya OPAL™ TSA fluorophores: OPAL 480, OPAL 570, AND OPAL 620 (Akoya Biosciences). Antibodies were validated by the providing companies and in this study using appropriate controls whenever possible. DAPI was used for nuclear counterstaining. Between each sequential antibody staining step, slides were incubated in citrate buffer pH 6 (Cell Conditioning Solution, Cat. #980-223, Roche Diagnostics) at 95 °C to remove the previous primary and secondary antibody complexes. Multiplex-stained slides were mounted with ProLong™ Gold Antifade Reagent (Cat. # P36934, ThermoFisher).

The entire slides were scanned at a 20x magnification using the Vectra® Polaris™ Automated Imaging System (Akoya Biosciences). Whole slide scans were reviewed using Phenochart™ whole slide contextual viewer software (Akoya Biosciences). Five representative images of 927 μm × 695 μm size were selected for each sample. Using the inForm® Software v2.4.10 (Akoya Biosciences), spectral unmixing and removal of auto-fluorescence were performed. Resulting images were segmented into individual cells and cells stained with different markers were quantified using the Phenotype algorithm in inForm software. Relative cell numbers were calculated by normalizing the number of each cell category by the sum of all relevant cell categories.

## Immunofluorescence staining and imaging on tissue sections

Embryos or adult pancreata were processed, sectioned, and deparaffinized as described for immunohistochemistry staining. Antigen retrieval was performed using antigen retrieval mix (Vector Laboratories, H-3300) in a pressure cooker for 15 min. 5% normal donkey serum + 0.5% Triton in PBS (PBST) was used to block for 1 h at room temperature. Primary antibodies raised in different species were added to the tissue in blocking solution overnight at 4 °C. After PBST washes, tissues were incubated in secondary antibodies conjugated with different fluorophores at room temperature for 2 h. After washes, slides were mounted with Fluoromount G. Zeiss confocal was used to image the tissues, followed with analysis in the Imaris software.

## Immunofluorescence staining and imaging with whole mount tissue

Embryos or pancreata were dissected and fixed with 4% PFA for 24 h at 4 °C. Tissues were then washed with PBS and dehydrated using a series of Methanol + PBS solutions. Tissues were permeabilized using Dent's Bleach (4:1:1 ratio of MeOH:DMSO:30% $H_2O_2$) for 2 h at room temperature. Tissues were rehydrated using a series of Methanol + PBS solutions. 5% normal donkey serum + PBST (blocking solution) was used to block for 2 h at room temperature. Primary antibodies were added to the tissue in blocking solution overnight at 4 °C. Five one-hour washes with 0.5% Triton in PBST were then performed at room temperature. Secondary antibodies diluted in blocking solution were added to tissues for overnight at 4 °C. Tissues were then washed three times of 30 min washes with PBST. Tissues were gradually dehydrated into MeOH and cleared in Murray's Clear (2:1 ratio of Benzyl Benzoate:Benzyle Alcohol). Zeiss confocal with Zen Black software was used to image the tissues, followed with Imaris analysis.

## Flow cytometry analysis and sorting

**Normal pancreas dissociation.** Dissociation media for normal pancreas was Collagenase P (1 mg/ml), Soybean Trypsin inhibitor (0.2 mg/ml), Dnase I (12 U/ml), HEPES (10 mM), Rock inhibitor Y (10 μM) in HBSS (with $Ca^{2+}$). The entire pancreas was dissected, minced in Eppendorf tubes, and then transferred to a Miltenyi gentleMACS C tube (Miltenyi Biotec, 130-093-237) with 2.5 ml of the dissociation media. Dissociation was performed on a gentleMACS Dissociator (Miltenyi Biotec, 130-093-235) using the 37m-TDK-1 program for 15 min.

**Tumor-bearing pancreas dissociation.** Dissociation media for tumor-bearing pancreas was prepared using the Tumor Dissociation Kit (Miltenyi Biotec, 130-096-730). Pancreas was dissected and cut into small pieces using a razor blade. A full program of 37m-TDK-1 was used to dissociate the tumor.

At the end of dissociation, normal and tumor samples were processed in the same way. 40 ml of cold DMEM + 10% fetal bovine serum (FBS) was then added to the dissociation mixture. Samples were then filtered through a 70 μm membrane and then a 100μm membrane. Red blood cell lysis buffer (Millipore Sigma, R7757) was then added to the pellet. The pellet was resuspended and pipetted gently for 1 min. 15 ml of PBS + 10%FBS was added to the sample. After centrifuge, cells are resuspended in FACS buffer (5% FBS, 10 μM Y, 20 mM HEPES, 5 mM $MgCl_2$, 12 U/ml DNase I in PBS). After Fc block, cells were incubated with fluorophore-conjugated antibodies (Supplementary Data 5).

BD FACS Aria machines were used to either analyze or sort cells using the FACS Diva software. Data were analyzed using Flow Jo. Side scatter (SSC) and forward scatter (FSC) were used to gate for cells, FSC (height) and FSC (area) were used to gate for single cells. DAPI or Zombie Aqua dye were used to negatively identify live cells. For fibroblast analysis, cells were selected as CD45-, CD31-, Epcam-, (Pdpn + or Pdgfrα +). GFP protein auto-fluorescence is detected using excitation: 488, emission: 530/30. Tomato protein auto-fluorescence is detected using excitation: 488, emission: 610/20. The percentage of certain cell types positive for certain reporters were calculated.

## Fibroblast culture from the pancreas

Pancreas was dissociated and processed following the same procedure as for flow analysis. Cells were plated into one well of a 12-well plate. Fibroblasts attached the following day and proliferated to establish cell lines. Cells were cultured in DMEM + 10% FBS + pen/strep media, which was refreshed every other day. Marker expression on cultured cells were examined using immunofluorescence staining similar to staining on tissue sections. Authentication of the generated cell lines were performed by immunostaining with known markers. Mycoplasma was not tested since only primary cells within five passages were used.

## Single-cell RNA sequencing

After dissociation and staining, cells were sorted into three groups, Group 1. hematopoietic cells (CD45 +); Group 2. endothelial (CD31 +) + epithelial cells (Epcam +); and Group 3. fibroblasts (Pdpn + or Pdgfrα + or Tomato +). For tumor samples, 4–5 mice were pooled to generate each replicate ($n = 2$). For normal samples, 6 mice were pooled, sorted and split into 2 replicates ($n = 2$). Cells were resuspended in PBS and counted with trypan blue to evaluate viability. To create a pool of hematopoietic cells (Group 1) and endothelial + epithelial cells (Group 2), an equal ratio of cells was mixed from these two groups to establish the pool 1 + 2.

Library preparation and sequencing were completed at the Medical College of Wisconsin Genomic Science and Precision Medicine Center with 10,000 cells aliquoted for groups 1 + 2 and 3. Samples were processed according to the Chromium Next GEM Single Cell 3' Reagent Kits (10X Genomics, dual indexes). All samples passed quality control metrics with 13–16 cycles of PCR completed to amplify the libraries. Final libraries were quantified and pooled by qPCR (Kapa Library Quantification Kit, Kapa Biosystems) and sequencing completed on the NovaSeq6000 targeting 50,000 reads per cell per condition.

## Single-cell RNA sequencing analysis

Sequenced datasets were processed in CellRanger and the output data was further analyzed in R using Seurat package. Genes expressed in less than 3 cells and cells with less than 200 genes were filtered out. nGene and percent.mito parameters were also used to remove multiplets and potentially dying cells. Global scaling was used to normalize counts across all cells in each sample and cell cycle effect was removed by regressing out genes differentiating between S phase and G2M phase. Approximately 2000 highly variable genes across each cluster were used to perform Principal Component Analysis (PCA). The first 20 PCA components were used for cell clustering, which were visualized using UMAP plots. Marker genes were identified using "FindAllMarkers" function in Seurat. This was used to identify differentially expressed (DE) genes between (1) normal fibroblasts and PDAC fibroblasts and (2) Tomato high clusters (0, 1, 13) and Tomato low cluster (6).

Cells from different samples across different studies were first normalized on a per-study base in Seurat, followed by prioritization of genes with more variable expression patterns, and then dataset merging using a nearest-neighbor approach using diagonalized canonical correlation analysis (CCA) subspaces for each pair of comparison groups[43]. All genome data of this study has been deposited in the NCBI Gene Expression Omnibus with the accession number GSE200903. For gene expression comparisons, we first performed integration of splanchnic mesenchyme (spl_meso[33]) and CAFs/TRFs dataset by using Pearson residuals (SCTransform with vst.flavor = "v2"). Then, three groups of differential expressed genes, including (1) spl_meso.vs.CAFs, (2) spl_meso.vs.TRFs, and (3) CAFs.vs.TRFs were identified using "FindMarkers" function in Seurat. Finally, genes highly expressed in both the splanchnic mesenchyme and CAFs compared to TRFs and genes highly expressed in both the splanchnic mesenchyme and TRFs compared to CAFs were identified by performing intersect function between three DE genes' group.

For human dataset analysis, we selected normal pancreatic fibroblasts from the Tabular Sapiens database[40] and PDAC fibroblasts from the FibroXplorer dataset[41]. We performed similar integration and gene expression comparisons as the mouse analysis. This identified genes highly expressed in both the splanchnic mesenchyme and human PDAC fibroblasts compared to human normal TRFs, and genes highly expressed in both the splanchnic mesenchyme and human normal TRFs compared to human PDAC CAFs. These DE genes were used to identify associated biological processes utilizing the NIH Gene Ontology resource (geneontology.org). Common biological processes were identified between the mouse and human analysis and the top six processes (based on their ranks in the human analysis) were shown. Splanchnic gene signatures were identified using "FindMarkers" in the foregut scRNA-seq dataset. The expression of these genes' orthologues were examined in human TRFs and CAFs.

## Reporting summary

Further information on research design is available in the Nature Portfolio Reporting Summary linked to this article.

## Data availability

Raw and processed mouse scRNA-seq data generated in this study have been deposited in the NCBI Gene Expression Omnibus (GEO) under accession code GSE200903. The publicly available human normal pancreatic scRNA-seq data used in this study are available in the Gene Expression Omnibus database under accession code GSE201333[40]. The publicly available mouse fetal foregut scRNA-seq data used in this study are available in the Gene Expression Omnibus database under accession code GSE136689[33]. The publicly available mouse PDAC scRNA-seq data used in this study are available in the Gene Expression Omnibus database under accession code GSE129455[10]. The raw data files are available from a public AWS S3 bucket (https://registry.opendata.aws/tabula-sapiens/)". The publicly available human PDAC sc-RNA seq data used in this study are available in the EGA database under accession EGAD00001005365. The integrated scRNA-seq objects used for analysis are provided in an online resource that can be accessed at https://fibroXplorer.com[41]. The remaining data are available within the Article, Supplementary Information or Source data file. Source data are provided with this paper.

## Code availability

The codes used in this study are available online at https://github.com/KunFang93/SplMeso_PDAC_NC. The code version used to generate data in this manuscript was deposited in zenodo (https://zenodo.org/record/7150239) and the corresponding DOI is as follows: https://doi.org/10.5281/zenodo.7150239[44].

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

## Acknowledgements

We thank members of the Ostrowski and the Leone laboratories, particularly C. Marelia and B. Bonala for technical assistance, and S. Sharma for advice and discussions related to experiments and analysis. We thank Dr. Helen Chamberlin for discussion of the project and feedback on the manuscript. We thank Elizabeth O'Quinn for technical assistance with slide imaging and analysis. We thank P. Chaturvedi and A. Zorn for sharing CellRanger processed dataset for the E9.5 foregut single-cell sample. We thank E. Elyada and D.A. Tuveson for sharing CellRanger processed dataset for the pancreatic cancer single-cell sample. This publication was supported in part by shared resources of Linda T. and John A. Mellowes Center for Genomic Sciences and Precision Medicine of Medical College of Wisconsin and Medical College of Wisconsin Cancer Center Shared Resources. We thank the following funders for fellowship and grant support: NIH/NCI Ruth L. Kirschstein NRSA Institutional Research Training Grant (T32CA193201, to L.H.), Ruth L. Kirschstein National Research Service Award for Individual Postdoctoral Fellows (F32CA254238, to L.H.), American Cancer Society – Fairfield County Comedy Against Cancer Committee Postdoctoral Fellowship in Cancer Research (PF-20-114-01-DDC, to L.H.), NCI P01 CA236778 (M.C.O.), NCI R01 CA258440 (X.Y.), NIAID R01 AI118305 (X.Y.), NIDDK R01 DK052913 (R.U.), the Advancing a Healthier Wisconsin Endowment (G.L. and A.H.W.), the John and Linda Mellowes Endowment (R.U.), and the Dr. Glenn R. and Nancy A. Linnerson Endowed Fund (G.W.L.).

## Author contributions

L.H., M.C.O., and G.W.L. designed and analyzed the experiments. L.H. and G.W.L. wrote the manuscript with contributions from all the authors. L.H. performed the majority of the experimental work. Y.W. assisted with the bone marrow transplantation experiments under the supervision of X.Y. S.S., U.K.R., and K.P. assisted with mouse genotyping. T.W. and J.P. performed tissue image analysis and quantification. M.P. performed all the histology processing and staining under the supervision of C.D.T. J.L. assisted with the c-sections of mice. A.T. assisted with tissue harvesting for single-cell RNA sequencing. A.J.M. performed single-cell RNA sequencing. M.T.Z., R.U., K.F., and V.X.J. performed single-cell RNA sequencing-related computational analysis. Critical reagents and resources were provided by X.Y., R.U., M.C.O., and G.L.

## Competing interests

The authors declare no competing interests.
