## [Peer Review File · Nature Communications]

The splanchnic mesenchyme is the tissue of origin for pancreatic fibroblasts during homeostasis and tumorigenesisReviewers' Comments:

Reviewer #1:

Remarks to the Author:

In this manuscript Han et al., describes the fetal-origin of CAFs in pancreatic cancers. They have employed state-of-art lineage tracking and single cell RNA-seq approaches to validate their findings. Overall this is a very interesting manuscript which links the oncofetal origin of stromal cells in cancer. This works, along with recently published work in liver cancer supports the oncofetal ecosystem concept. Overall this is a very good work and provides novel insights into phylogeny of CAFs in pancreatic cancer. Authors have done elegant experiments; however, I have only one major concern/request before this work is acceptable for publication in Nature Communications. This work will be more relevant if authors can probe the origin of CAFs in human PDACs by integrating fetal pancreas and tumor data. I think this analysis/integration will provide novel insights into biology of human cancers and pave the way for clinical implications of this discovery.

Reviewer #2:

Remarks to the Author:

In this study, Han et al. report that the splanchnic mesenchyme is the tissue of origin of pancreatic fibroblasts in homeostasis and tumorigenesis. It is an interesting subject to the pancreatic research community and the manuscript is well- and clearly written. The origin of the fibroblasts that constitute the desmoplastic milieu surrounding pancreatic tumors is a subject of active discussion. In addition, studies have characterized the cancer associated fibroblasts (CAFs) surrounding pancreatic cancer into myofibroblast and inflammatory subtypes – both believed to contribute to tumor progression and biology. Importantly, the pancreatic fibroinflammatory environment is considered a barrier to drug delivery and therapy. Thus, understanding the progenitors of the pancreatic fibroblasts could expose new opportunities for improving the tumor treatment. Overall, the manuscript is interesting.

Main comments

1. The use of only one single marker 'Isl1' to identify the mesenchyme is really concerning in terms of rigor and tends to assume (just based on one prior publication) that Isl1 is well established that Isl1 is a marker of the splanchnic mesenchyme. This gene, according to the page describing the mouse model used in this study (<https://www.jax.org/strain/024242>) "is expressed in undifferentiated cardiac progenitors and hindlimb progenitors". Is it established that this mouse model is used for tracing fibroblast origin?
2. Following up on the above, it appears Isl1 was first mentioned in line 100 – not introduced and no concrete rationale for focusing the bulk of the study on it as the sole marker of splanchnic mesenchyme surrounding the pancreas. In my view, this is a major weakness of this study. Authors should identify other markers that would help support that the mesenchyme were indeed specifically marked by Isl1 (especially considering the UMAP in Fig. 2b, where both the ectoderm and endoderm were also marked by Isl1).
3. In the experiment with bone marrow transplantation, bone marrow cells from the UBC-GFP+ donor mice were transplanted into the KPF mice by tail vein injection. Besides enriching the circulation with these cells (to mimic the idea of migration to the pancreas) is there a reason why the mice were not crossbred rather than tail vein injection? The concern here is that injections by tail vein often target the liver. Therefore, I wouldn't be too surprised that the injected cells (were more lodged in the liver) and so did not end up in the pancreas. This raises concern on the strength of the conclusion drawn based on this very tail vein injection experiment.
4. The authors report that Isl1+ mesenchyme gives rise to tissue resident fibroblasts (TRFs) and CAFs. Are these simultaneous or sequential events? In other words, does the Isl1+ mesenchyme give rise to

TRFs that subsequently give rise to CAFs, or does CAFs still emerge directly from Isl1+ mesenchyme even when TRFs are present?

5. It seems to me that it would be worthwhile to leverage on the single cell RNA sequencing (scRNA seq) data to delineate gene expression differences (if any) between the splanchnic mesenchyme-derived fibroblasts in the normal versus tumors. So far, besides few UMAPs, not much was learned from the scRNA seq data.

Minor comments

1. In line 16, the authors wrote that "Previous work showed that some pancreatic tumor cells express fibroblast markers, raising the possibility that tumor cells have the potential to give rise to CAFs". This is a rather unnecessary and 'likely-to-be-wrongly-interpreted' speculation. The mere expression of signatures of CAFs by tumors does not imply that those CAFs may have arisen from tumors. This speculation should be removed.

2. Provide a table of the gene signatures represented in Fig 5f. Are these genes all similarly upregulated or downregulated? For instance, are the 473 genes identified to overlap between E9.5 Spl_Meso and PDAC fibroblasts all high or low in both cell types? The genes should be separated into highly- or lowly expressed subsets between the cell types. This will also enable the identification of surrogates to Isl1 supposing it is confirmed that it is specific for the splanchnic mesenchyme.

3. The schematic in 5a should be simplified by providing further (perhaps 1 sentence) description in the legend. In the current state it is hard to decipher what message the schematic conveys.

Reviewer #3:

Remarks to the Author:

This is rigorous study using at least three different lineage tracing models (two autochthonous, and one of transplantation) that establishes the splanchnic mesenchyme as the major source of tissue resident fibroblasts (TRFs) in the normal pancreas, and CAFs in pancreatic cancer. This study add to the burgeoning story of CAF origins in PDAC, filling a major piece of the puzzle.

My only question would be to please provide some human data - surely the investigators can go back to patient scRNA datasets that are published from both neoplastic and normal pancreas (including the recently published tabula sapiens database) and mine for the splanchnic mesenchyme "primitive" signature in those settings. That would complete the story.

Reviewer #4:

Remarks to the Author:

The origin/source of cancer-associated fibroblasts (CAFs) in pancreatic cancer is unclear and therefore been debated for a long time. In recent years, lineage tracing experiments have tried to resolve this, and it seems that the stellate cells, that in the past been claimed to be a major source of CAFs, might contribute less than previously thought.

Here, the authors use lineage tracing and single cell transcriptomics to reveal the fetal splanchnic mesenchyme as an important source of fibroblasts in the pancreas and in pancreatic cancer. This provides important insights on the source of CAFs in pancreatic cancer.

General comments:

- No mention is made (for instance in Lines 135-153) if the constitutive Isl1Cre reporter for the splanchnic mesenchyme gives rise to fibroblasts in the lung, liver, stomach and gut. If this has been observed, it could be useful to readers interested in these organs. Likewise, on line 116 and fig 2c, it is not clear whether isl1 expression is restricted to the mesenchyme around the pancreas or whether it is also found around other endoderm-derived organs. Could the authors comment on that?

- Recently, a CD105- pancreatic fibroblast lineage has been suggested to support anti-tumor immunity (PMID: 34297917). What about the status of CD105 in the splanchnic lineage fibroblasts? Would be relevant to discuss this in the manuscript.

- The definition used for CAFs in this study is: "cells located in the stroma or interstitium, positive for markers including VIMENTIN (VIM), PDGFR α , PDGFR β , PDPN or α SMA, and negative for other lineage markers". With this definition EMT cells can be considered to be CAFs. In my opinion, a more accepted definition for CAFs is: "all the fibroblastic, non-neoplastic, non-vascular, non-epithelial and non-inflammatory cells found in a tumor ", i.e., according to this definition, neoplastic EMT cells cannot be CAFs. I would suggest that the later definition is used, and that the text related to figure 1 is changed so that EMT cancer cells are not referred to as CAFs.

- In the single cell seq experiment (figure 5), not all fibroblasts are tomato positive. Some myofibroblasts and a quite large proportion of inflammatory fibroblasts (fig 5d) are tomato negative. What are these cells? Please elaborate on this. Would it be possible to look in detail between the differences tomato+ and tomato- cells in the fibroblast clusters to get an idea of the origin of the tomato- fibroblasts?

Minor and specific comments:

- Please elaborate on why the transcription factor Isl1 was identified and used in this study.

- A model where p53 is deleted was used in this study. Is there a reason why this was used instead of a model with point mutations in p53?

- Line 81: "This analysis showed that while more than 80% of EpCAM+ epithelial cells were labeled with GFP, very few PDPN+ or PDGFR α + CAFs". The GFP+ cells could be doublets of epithelial and fibroblast cells. Or fibroblasts that have incorporated material originating from tumour cells (transfer of mRNA or proteins). In that respect, if PDGFR α were co-stained, it would be useful to gate the GFP+ cells and show a plot with EpCAM vs PDGFR α (could be in supplementary material) .

- If the data herein allows, it would be very interesting to examine whether any genes expressed in embryonic splanchnic mesenchyme are downregulated in pancreas resident fibroblasts but reactivated in CAFs.

- Line 193: "a finding of significant relevance to pancreatic morphogenesis and diseases to which these cell types contribute". A reference to the literature showing involvement of iCAF and myCAF in pancreatic morphogenesis is missing.

- It seems that very few EMT cells in the stroma (Figure 1). How is this finding in relation to previous findings using similar animal models (Rhim et al, PMID: 22265420)? Please discuss about this.

- In Fig 1b-c it is difficult to see any double positive cells. Could a representative image containing areas with double positive cell be included?

- - PDGFR α and PDGFR β are used differently (interchangeable) in flow and IHC. Why is not the same

marker used in both types of experiments? The different receptors might be markers for different fibroblast populations.

Reviewer #5:

Remarks to the Author:

Han et al provide fundamental evidence that the majority of fibroblasts residing in normal pancreas or PDAC come from the same origin, the *Isl1*-expressing splanchnic mesenchyme. This is significant because, despite the widespread recognition of multiple subsets of fibroblasts with different functions, the source of heterogeneity has yet to be elucidated. Given that this article revealed that the majority of CAFs is derived from the same origin, different phenotypes of fibroblasts might be dependent on the microenvironmental cues, not their original sources.

The authors designed complete and thorough experimental plans, and the manuscript is well-written; the results and potential meaning of this article is significantly novel. There are only some minor details that the authors need to validate or to state clearly for readers to understand them well through the figures.

Specific comments:

1. IcreT model and KPFIcreT model

1) In Figure 2n and Fig 3l, relatively small amounts of Tomato+ cells still exist in CD45+, EpCAM+, CD31+ cells, suggesting that these cell types could also have originated from the splanchnic mesenchyme. I wonder whether there are previous reports supporting this result, or is it just a non-specific fluorescence signal observed in these mouse models?

2) I am really impressed that the authors applied the inducible cre model to exclusively label fibroblast, not epithelial and hematopoietic cells. However, I'm not sure why the injection into E9.5, rather than E8.5 or 10.5, could only label fibroblasts with Tomato. Are the authors able to make any speculations?

2. Single cell data

1) I wonder whether the authors confirmed the different subtypes of fibroblasts in between IcreT and KPFIcreT tissues. I can make an estimate based on Fig 5b, but the more thorough presentation will allow readers to compare the findings to past studies.

2) The authors stated that they found both inflammatory fibroblasts and myofibroblasts in *Isl1*+ splanchnic mesenchyme-derived cells. But I wonder whether authors could detect both subtypes using tissue staining. Additional experiments with different methods would solidify their findings. There are many papers which identified and quantified the different types of CAFs in PDAC based on their markers.

-Distinct populations of inflammatory fibroblasts and myofibroblasts in pancreatic cancer, *J Exp Med*. 2017 Mar 6;214(3):579-596. doi: 10.1084/jem.20162024. Epub 2017 Feb 23

-IL-1-induced JAK/STAT signaling is antagonized by TGF- β to shape CAF heterogeneity in pancreatic ductal adenocarcinoma. *Cancer Discov*. 2019;9:282-301.

-Differential Contribution of Pancreatic Fibroblast Subsets to the Pancreatic Cancer Stroma, *Cell Mol Gastroenterol Hepatol*. 2020; 10(3): 581-599. Published online 2020 May 23. doi: 10.1016/j.jcmgh.2020.05.004

3) I agree with the authors' statement that there might be common regulatory mechanisms within the splanchnic lineage throughout development, homeostasis and cancer, but I can't see the potential importance of this finding. I wonder whether authors could further explain these results.

4) Fig5a,b: Given that IcreT and KPFIcreT models have Tomato+ signals in non-fibroblasts cells such as EpCAM+, CD45+, CD31+, I wonder why the authors chose these models rather than inducible cre models for the single cell analysis. I think there could be EpCAM, CD45, CD31 expressions within the

fibroblasts plot of Fig. 5b.

5) Since the authors performed the single-cell transcriptomic analyses, I wonder If the authors could detect the different genomic signatures within a certain cell type which possibly affects the CAF heterogeneity in PDAC.

Dear reviewers,

We greatly appreciate the comments and feedback provided. We carefully considered all the comments and now include additional data, rationale where requested, and clarification. Beyond reviewer requests, we also added additional marker analysis and quantifications to corroborate the original conclusions, including Fig. 4 a, b, i and Extended Data Fig. 1d. The text and figures were adjusted accordingly. We believe these revisions have added clarity and strengthened the conclusions. A detailed response to each of the reviewer comments are detailed below.

Reviewer #1:

“This work will be more relevant if authors can probe the origin of CAFs in human PDACs by integrating fetal pancreas and tumor data.”

We obtained published single cell RNA sequencing datasets of pancreata from normal human and PDAC human patients. We compared the gene expression profiles of human pancreatic fibroblasts to the fetal mouse splanchnic mesenchyme. This identified a group of genes highly expressed only in the mouse splanchnic mesenchyme and human normal fibroblasts, and another group of genes highly expressed in only the mouse splanchnic mesenchyme and human PDAC CAFs (new Fig. 5f, g, new Extended Data Table 4). These mouse and human gene expression comparisons suggest that CAFs in human PDAC may, like our mouse lineage tracing experiments demonstrate, potentially originate from the splanchnic mesenchyme. Additionally, we found that fetal splanchnic mesenchyme gene signatures are also expressed in both human normal fibroblasts and human PDAC CAFs (new Extended Data Fig. 7j), suggesting that expression of a common signature may be involved in maintaining their cell lineage identity. This latter hypothesis will require extensive future experimentation to be tested and have thus, not included in the manuscript text.

Reviewer #2:

Main Comments

“1. The use of only one single marker ‘Isl1’ to identify the mesenchyme is really concerning in terms of rigor and tends to assume (just based on one prior publication) that Isl1 is well established that Isl1 is a marker of the splanchnic mesenchyme. This gene, according to the page describing the mouse model used in this study (<https://www.jax.org/strain/024242>) “is expressed in undifferentiated cardiac progenitors and hindlimb progenitors”. Is it established that this mouse model is used for tracing fibroblast origin?”

All cell fate mapping is biased by the gene used as the driver. We also acknowledge that ISL1 is also expressed in cardiac and limb progenitors, however certain cardiac progenitors reside in the second heart field, which are common progenitors for both the heart and the foregut mesenchyme (Peng, 2013, *Nature*).

ISL1 is well established as a marker of the general splanchnic mesenchyme. We have now added more rationale including the citation of publications (a total of 5) in line 145-147. These authors showed that ISL1 is expressed in the mesenchyme surrounding the foregut endoderm (i.e. the splanchnic mesenchyme). Additionally, ISL1 is co-expressed with other splanchnic markers including Tbx5 and Gata6 (Rankin and Han, 2016, *Cell Reports*). In line 149-150, we added rationale that *Isl1^{cre}* has already been used to lineage trace the origin of the stromal components of foregut derived organs (lung and esophagus).

In this current study, we experimentally confirmed that ISL1 is indeed expressed in the splanchnic mesenchyme surrounding the fetal pancreas. Therefore, we hypothesized *Isl1^{cre}* and *Isl1^{creER}* can target the splanchnic mesenchyme. Tomato reporter expression in the splanchnic mesenchyme at E12.5 supports this hypothesis. We used four mouse models, including *Isl1^{cre}* and *Isl1^{creER}*, to

identify CAF origin and exclude other potential CAF tissue sources. The comprehensive analysis using multiple mouse models to include or exclude potential sources from which pancreatic CAFs arise is well beyond what is normally expected in the field, and thus, we feel confident with the conclusions drawn in this manuscript.

“2. Following up on the above, it appears *Isl1* was first mentioned in line 100 – not introduced and no concrete rationale for focusing the bulk of the study on it as the sole marker of splanchnic mesenchyme surrounding the pancreas. In my view, this is a major weakness of this study. Authors should identify other markers that would help support that the mesenchyme were indeed specifically marked by *Isl1* (especially considering the UMAP in Fig. 2b, where both the ectoderm and the endoderm were also marked by *Isl1*).”

Thank you for noticing this oversight. Additional rationale is now included to support the use of *Isl1* to lineage trace the splanchnic mesenchyme (details see point 1 above).

We used a total of five markers to identify fibroblasts by immunostaining and flow cytometry analysis. We found that progenitor cells targeted by *Isl1^{cre}* and *Isl1^{creER}* gave rise to approximately 70-90% of fibroblasts in the normal pancreas and in PDAC (Fig 4m, Extended Data Fig. 5d-f). Single cell RNA sequencing shows the co-expression of Tomato with well-established fibroblast markers within the same cluster, supporting our conclusions. We are confident that the pancreatic mesenchyme is indeed targeted by *Isl1^{cre}* and *Isl1^{creER}*.

We agree that Fig. 2b shows *Isl1* expression in some endoderm and ectoderm cell populations. Limiting *creER* activity with one gavage of Tamoxifen resulted in further restricting *creER* activity to the mesenchyme, with very few endodermal cells expressing the Tomato reporter, demonstrated by flow cytometry data in Fig. 4m (<0.6% of EPCAM+ epithelium was labeled with Tomato) and the new immunostaining quantification in Fig. 4i (<0.1% of ECAD+ epithelium was

labeled with Tomato). Ectoderm derived cells in the pancreas are rather sparse and thus not investigated in this study.

“3. In the experiment with bone marrow transplantation, bone marrow cells from the UBC-GFP+ donor mice were transplanted into the KPF mice by tail vein injection. Besides enriching the circulation with these cells (to mimic the idea of migration to the pancreas) is there a reason why the mice were not crossbred rather than tail vein injection? The concern here is that injections by tail vein often target the liver. Therefore, I wouldn't be too surprised that the injected cells (were more lodged in the liver) and so did not end up in the pancreas. This raises concern on the strength of the conclusion drawn based on this very tail vein injection experiment.”

Bone marrow transplantation through venous infusion is a well-established system (greater than 10,000 citations) for bone marrow engrafting in both mice and humans (Nguyen, 2020, Clin Cancer Res; Duran-Struuck, 2009, JAALAS; Watanabe, 2009, Am J Physiol Gastrointest Liver Physiol; pancreatitis: Marrache, 2008, gut; liver fibrosis: Mederacke, 2013, Nat Commun; and our own work: Bazinet, 2019, Curr Oncol). Even though injection by tail vein often targets the lung and the liver through the natural circulation, “homing” of injected cells to the bone marrow is dependent on specific homing factors such as stromal-derived factor 1 and stem cell factor. Therefore, delivery of cells into the circulation allows bone marrow cells to establish residence in the bone marrow of the new host, hence, lethally irradiated mice used in many hematopoietic cell reconstitution studies survive when transplanted by this method. Citations supporting this method is now included in the text (line 117-118).

To demonstrate successful bone marrow engraftment in our current study, we now include additional data and description in the text (line 117-119): 1. Irradiated mice that didn't receive bone marrow died within 10 days and mice that received a bone marrow transplant by tail vein injection lived up to a year(n>6 mice for each group, data not included); 2. Peripheral blood flow

cytometry analysis five months after injection showed, as expected, that >85% of blood cells were GFP+ (Extended Data Fig. 1u, newly added). The long-term mouse survival and persistence of blood cells would not have been achieved without successful bone marrow engrafting. Moreover, in our current study, there were abundant GFP+ cells in the pancreas (immunostaining data on tissue section Fig. 1j-k, and the number of GFP+ cells were as abundant as fibroblasts in the tumor-bearing pancreas Fig. 1l).

“4. The authors report that *Isl1*+ mesenchyme gives rise to tissue resident fibroblasts (TRFs) and CAFs. Are these simultaneous or sequential events? In other words, does the *Isl1*+ mesenchyme give rise to TRFs that subsequently give rise to CAFs, or does CAFs still emerge directly from *Isl1*+ mesenchyme even when TRFs are present?”

We agree with the cautionary note. We have now softened the language in the discussion (line 304-305). Currently we cannot firmly distinguish the two possibilities raised by the reviewer. We speculate that these are sequential events based on the following reasons: 1. When KPFIT mice were harvested at a younger age (~30 days old mice), we observed a localized and acute expansion of fibroblasts surrounding early lesions (H&E staining not shown), while fewer fibroblasts (presumably TRFs) were scattered across the adjacent normal pancreatic tissue. All of these fibroblasts (in the tumor and non-tumor areas) were labeled with Tomato expression (Fig. 3d and newly added Extended Data Fig. 3g-h), suggesting a continuous presence of splanchnic derived fibroblasts during the entire process of tumorigenesis. 2. Other recent studies show that certain subtypes of TRFs in the pancreas can give rise to CAFs in PDAC (Garcia, 2020, *Cell Mol Gastroenterol Hepatol*; Helms, 2020, *Cancer Discov*). Based on these two observations, we believe it is a logical extension to suggest that the splanchnic mesenchyme gives rise to TRFs first, which then expand during tumorigenesis to form CAFs.

“5. It seems to me that it would be worthwhile to leverage on the single cell RNA sequencing (scRNA seq) data to delineate gene expression differences (if any) between the splanchnic mesenchyme-derived fibroblasts in the normal versus tumors. So far, besides few UMAPs, not much was learned from the scRNA seq data.”

A heatmap of differentially expressed genes between normal fibroblasts and PDAC fibroblasts is now included as Extended Data Fig. 7e. A spreadsheet of gene lists is provided as Extended Data Table 2. While beyond the scope of this study, this analysis will foster further investigation into gene expression changes in fibroblasts and their functional consequences during tumorigenesis.

“Minor comments

1. In line 16, the authors wrote that “Previous work showed that some pancreatic tumor cells express fibroblast markers, raising the possibility that tumor cells have the potential to give rise to CAFs”. This is a rather unnecessary and ‘likely-to-be-wrongly-interpreted’ speculation. The mere expression of signatures of CAFs by tumors does not imply that those CAFs may have arisen from tumors. This speculation should be removed.”

We have removed this speculation. See line 91.

“2. Provide a table of the gene signatures represented in Fig 5f. Are these genes all similarly upregulated or downregulated? For instance, are the 473 genes identified to overlap between E9.5 Spl_Meso and PDAC fibroblasts all high or low in both cell types? The genes should be separated into highly- or lowly expressed subsets between the cell types. This will also enable the identification of surrogates to Isl1 supposing it is confirmed that it is specific for the splanchnic mesenchyme.”

The numbers in the original Fig. 5f indicate the relative distance between each cell type based on gene expression similarity. We understand how this figure is difficult to interpret, and thus, to improve clarity and provide readers with meaningful gene identities, we have now replaced the original Fig. 5e, f with an updated Fig. 5e showing a heatmap of differentially expressed genes comparing E9.5 splanchnic mesenchyme, PDAC fibroblasts and normal fibroblasts. As suggested by the reviewer, we separated the genes into highly or lowly expressed subsets between the cell types. The lists of genes are also provided as Extended Data Table 3.

“3. The schematic in 5a should be simplified by providing further (perhaps 1 sentence) description in the legend. In the current state it is hard to decipher what message the schematic conveys.”

We have simplified Fig. 5a and added more description in the legend (line 770-773).

Reviewer #3:

“My only question would be to please provide some human data - surely the investigators can go back to patient scRNA datasets that are published from both neoplastic and normal pancreas (including the recently published tabula sapiens database) and mine for the splanchnic mesenchyme "primitive" signature in those settings. That would complete the story.”

As mentioned in our response to reviewer 1, we obtained publicly available single cell RNA sequencing datasets of pancreata from normal human and pancreatic cancer human patients, including the suggested tabular sapiens database. We found that the splanchnic mesenchyme gene signatures are indeed expressed in tissue resident fibroblasts (TRFs) from the normal pancreata and cancer associated fibroblasts (CAFs) from the PDAC pancreata, with certain genes expressed at similar levels, certain genes expressed higher in TRFs and certain genes expressed higher in CAFs. This data is now added as Extended Data Fig. 7j.

Reviewer #4:

General comments:

“- No mention is made (for instance in Lines 135-153) if the constitutive *Isl1Cre* reporter for the splanchnic mesenchyme gives rise to fibroblasts in the lung, liver, stomach and gut. If this has been observed, it could be useful to readers interested in these organs. Likewise, on line 116 and fig 2c, it is not clear whether *isl1* expression is restricted to the mesenchyme around the pancreas or whether it is also found around other endoderm-derived organs. Could the authors comment on that?”

We intend to prepare a separate manuscript to fully characterize this model, which will include a comprehensive description of how this *Isl1Cre* reporter allele could be used to target the pancreas and other organs. We have provided part of the data below for the purpose of responding to this comment (not in manuscript). In E12.5 embryos, *Isl1cre* directed Tomato expression is evident in some of the mesenchyme around the lung and the stomach (Fig. a, b, n=2). In adult mice, *Isl1cre* directed Tomato is also expressed in some stromal cells in the stomach and intestine (Fig. c, d, n=2).

“- Recently, a CD105- pancreatic fibroblast lineage has been suggested to support anti-tumor

immunity (PMID: 34297917). What about the status of CD105 in the splanchnic lineage fibroblasts? Would be relevant to discuss this in the manuscript.”

CD105 (encoded by *Eng*) is expressed in a subset of the splanchnic lineage fibroblasts. This data is now added as Extended Data Fig. 7d.

“- The definition used for CAFs in this study is: “cells located in the stroma or interstitium, positive for markers including VIMENTIN (VIM), PDGFR α , PDGFR β , PDPN or α SMA, and negative for other lineage markers”. With this definition EMT cells can be considered to be CAFs. In my opinion, a more accepted definition for CAFs is: “all the fibroblastic, non-neoplastic, non-vascular, non-epithelial and non-inflammatory cells found in a tumor “, i.e., according to this definition, neoplastic EMT cells cannot be CAFs. I would suggest that the later definition is used, and that the text related to figure 1 is changed so that EMT cancer cells are not referred to as CAFs.”

Defining what is a fibroblast continues to be a challenge and remains a contested area of investigation. While we appreciate the suggestion made by the reviewer, a significant part of the study is to rigorously test whether epithelium is a source of CAFs. Thus, we believe a definition of CAFs based on marker expression, rather than on the presumed origin (such as non-neoplastic cells), would be more appropriate in this study, which is aimed at identifying the tissue source of CAFs. Our definition, while broad in nature, will avoid confusion and aid future comparative studies by us and others to be properly evaluated. Even with such a broad definition of CAFs, epithelium derived cells do not meet the criteria.

“- In the single cell seq experiment (figure 5), not all fibroblasts are tomato positive. Some myofibroblasts and a quite large proportion of inflammatory fibroblasts (fig 5d) are tomato negative. What are these cells? Please elaborate on this. Would it be possible to look in detail

between the differences tomato+ and tomato- cells in the fibroblast clusters to get an idea of the origin of the tomato- fibroblasts?”

Thank you for pointing this out. We agree that a proportion of inflammatory fibroblasts showed lower level of Tomato signal. This negative data could be due to several reasons including: 1. They do not arise from the splanchnic mesenchyme and thus, have a different cell origin. 2. Technical reasons, which may include the unlikely scenario where 100% of progenitors cells are label/marked by Cre/CreER, and/or the potential gene “drop-out” rate due to shallow sequencing depth with single cell RNA sequencing technique, among other possibilities. However, bone marrow and epithelium lineage tracing showed minimal contribution to CAFs. Because of these limitations, we feel uncomfortable making further conclusions based on this type of negative data. To aid further investigation in the future, we chose to simply describe the observation (line 260-261) and include the list of genes differentially expressed between Tomato low vs. Tomato high cell clusters as Extended Data Table 1, without making any hard conclusion based on this negative data.

Minor and specific comments:

“- Please elaborate on why the transcription factor *Isl1* was identified and used in this study.”

In line 145-147 and line 149-150, we added further rationale and citations (a total of 5) supporting the use of *Isl1* gene as a targeting driver for cell lineage tracing experiments. *ISL1* expression is well established as a marker of the general splanchnic mesenchyme. *Isl1^{cre}* has also been used previously to lineage trace the origin of the stromal components of foregut derived organs (lung and esophagus). We further confirmed that *Isl1* is indeed expressed in the splanchnic mesenchyme surrounding the fetal pancreas. Therefore, we hypothesized that *Isl1^{cre/creER}* can target the splanchnic mesenchyme.

“- A model where p53 is deleted was used in this study. Is there a reason why this was used instead of a model with point mutations in p53?”

Both p53 deletion and point mutations are present in PDAC human patients and both versions are widely used in generating PDAC mouse models (Rhim, 2012, *Cell*; Hingorani, 2005, *Cancer Cell*). Both versions recapitulate main characteristics of PDAC with mild differences in CAF characteristics (Flowers, 2021, *Cancer Discov*). However, to be compatible with the *Pdx1^{Flo}* allele generated in our laboratory, we opted to use the p53 frt allele, which we have used previously to result in PDAC formation (Wu, 2017, *PLoS One*). To our knowledge, p53 frt point mutation alleles have not been generated.

“- Line 81: “This analysis showed that while more than 80% of EpCAM+ epithelial cells were labeled with GFP, very few PDPN+ or PDGFR α + CAFs”. The GFP+ cells could be doublets of epithelial and fibroblast cells. Or fibroblasts that have incorporated material originating from tumour cells (transfer of mRNA or proteins). In that respect, if PDGFR α were co-stained, it would be useful to gate the GFP+ cells and show a plot with EpCAM vs PDGFR α (could be in supplementary material) .”

PDGFR α was co-stained with EpCAM and GFP. As suggested, we gated the GFP+ cells and showed a plot with EpCAM vs PDGFR α , which is now included as Extended Data Fig. 1k. This analysis showed that more than 97% of GFP+ cells were EPCAM positive while doublets were minimally present (<1%).

“- If the data herein allows, it would be very interesting to examine whether any genes expressed in embryonic splanchnic mesenchyme are downregulated in pancreas resident fibroblasts but reactivated in CAFs.”

We now added this data as Fig. 5e and Extended Data Table 3.

“- Line 193: “a finding of significant relevance to pancreatic morphogenesis and diseases to which these cell types contribute”. A reference to the literature showing involvement of iCAf and myCAF in pancreatic morphogenesis is missing.”

We are not aware of studies showing involvement of iCAF and myCAF in morphogenesis, thus “morphogenesis” is now removed from the text (see line 260). We appreciate this input to increase accuracy of our manuscript.

“- It seems that very few EMT cells in the stroma (Figure 1). How is this finding in relation to previous findings using similar animal models (Rhim et al, PMID: 22265420)? Please discuss about this.”

Discussion about this point is now included in the text (line 101-102). This could be due to different models of PDAC, where the Rhim study utilizes the cre/loxP system and our current study utilizes the FlpO/frt system, or due to different genetic backgrounds of the mice used in the two studies. For example, Rhim et al report a high incidence of metastasis to the liver, whereas we rarely observe metastasis. To note, the observation that GFP positive epithelial tumor cells could be found in the stroma in the Rhim et al study were not quantified.

“- In Fig 1b-c it is difficult to see any double positive cells. Could a representative image containing areas with double positive cell be included?”

In these mouse models, barely any double positive cells can be identified as quantified in Fig. 1d. Therefore, we believe the current images without any double positive cells are appropriate and representative images.

“- - PDGFRa and PDGFRb are used differently (interchangeable) in flow and IHC. Why is not the same marker used in both types of experiments? The different receptors might be markers for different fibroblast populations.”

This was designed based on availability of commonly used antibodies for different assays. Both PDGFR α and PDGFR β showed rather similar results, adding rigor in our conclusion.

Reviewer #5:

1. IcreT model and KPFIcreT model

“1) In Figure 2n and Fig 3l, relatively small amounts of Tomato+ cells still exist in CD45+, EpCAM+, CD31+ cells, suggesting that these cell types could also have originated from the splanchnic mesenchyme. I wonder whether there are previous reports supporting this result, or is it just a non-specific fluorescence signal observed in these mouse models?”

There are no previous reports describing this observation. We believe that the small amounts of targeting is real instead of non-specific fluorescence because: 1. This was absent in the Tomato negative samples; 2. This could be avoided using *Isl1^{creER}*, as shown in Fig. 4m.

“2) I am really impressed that the authors applied the inducible cre model to exclusively label fibroblast, not epithelial and hematopoietic cells. However, I'm not sure why the injection into E9.5, rather than E8.5 or 10.5, could only label fibroblasts with Tomato. Are the authors able to make any speculations?”

We speculate that a dynamic expression of *Isl1* during development may underlie this observation. At E8.5, *Isl1* is expressed in certain endoderm progenitors (potentially the ventral foregut) that preferentially give rise to epithelial cells in the head of the pancreas; at E9.5, such

endodermal progenitor expression is repressed; at E10.5, as certain endodermal progenitors are being specified to form islet cells, *Isl1* begins to be expressed in these cells.

2. Single cell data

“1) I wonder whether the authors confirmed the different subtypes of fibroblasts in between IcreT and KPFIcreT tissues. I can make an estimate based on Fig 5b, but the more thorough presentation will allow readers to compare the findings to past studies.”

In addition to the original Extended Data Fig. 7b, we now present the fibroblast clusters split into either IcreT or KPFIcreT samples (Extended Data Fig. 7c).

“2) The authors stated that they found both inflammatory fibroblasts and myofibroblasts in Isl1+ splanchnic mesenchyme-derived cells. But I wonder whether authors could detect both subtypes using tissue staining. Additional experiments with different methods would solidify their findings. There are many papers which identified and quantified the different types of CAFs in PDAC based on their markers.

-Distinct populations of inflammatory fibroblasts and myofibroblasts in pancreatic cancer, *J Exp Med.* 2017 Mar 6;214(3):579-596. doi: 10.1084/jem.20162024. Epub 2017 Feb 23

-IL-1-induced JAK/STAT signaling is antagonized by TGF- β to shape CAF heterogeneity in pancreatic ductal adenocarcinoma. *Cancer Discov.* 2019;9:282–301.

-Differential Contribution of Pancreatic Fibroblast Subsets to the Pancreatic Cancer Stroma, *Cell Mol Gastroenterol Hepatol.* 2020; 10(3): 581–599. Published online 2020 May 23. doi: 10.1016/j.jcmgh.2020.05.004”

As suggested, we used tissue staining to detect both subtypes of fibroblasts. As described in *J Exp Med.* 2017 report, we used high α SMA staining to identify myofibroblasts and low α SMA

staining to identify inflammatory fibroblasts. Consistent with our scRNA data, both subtypes are labeled with Tomato expression in our KPFI model. This data is now included as Extended Data Fig. 7f-i.

“3) I agree with the authors’ statement that there might be common regulatory mechanisms within the splanchnic lineage throughout development, homeostasis and cancer, but I can’t see the potential importance of this finding. I wonder whether authors could further explain these results.” Whether expression of key genes is persistent or is re-engaged in disease (cancer) may provide insight into their functions and could potentially be used therapeutically. Moreover, these findings could be used to identify common regulatory mechanisms that determine persistent versus re-engaged gene expression patterns, providing the identity of additional factors that may differentiate normal versus cancer cells.

“4) Fig5a,b: Given that IcreT and KPFIcreT models have Tomato+ signals in non-fibroblasts cells such as EpCAM+, CD45+, CD31+, I wonder why the authors chose these models rather than inducible cre models for the single cell analysis. I think there could be EpCAM, CD45, CD31 expressions within the fibroblasts plot of Fig. 5b.”

This was based on pragmatic and logistical reasons. The *KPFI^{creERT}* model is rather complex to generate, involving timed mating, tamoxifen gavage and C-section, making it logistically difficult to coordinate the generation of such a large cohort of mice at a single time point.

“5) Since the authors performed the single-cell transcriptomic analyses, I wonder If the authors could detect the different genomic signatures within a certain cell type which possibly affects the CAF heterogeneity in PDAC.”

This is an interesting question, but the current system is not well suited to address this due to several reasons:

- 1) Biology: These samples were pooled from multiple mice, therefore, it's impossible to correlate mutations in any tumor cells to their associated fibroblasts. Furthermore, genetic mutations within CAFs are very unlikely.
- 2) Bioinformatics: Data generated using 10x scRNA-seq platform is inherently associated with low transcript abundance, allelic dropout, and incomplete transcript coverage. To call variants, we need to have sufficient coverage of the bases, which is not attainable by the methods used in the current study. Thus, estimation of genome-wide mutations is naturally limited and could be potentially misleading.

Reviewers' Comments:

Reviewer #1:

Remarks to the Author:

The authors have addressed all the concerns raised during peer review. I would like to recommend the acceptance of this manuscript.

Reviewer #2:

Remarks to the Author:

Many thanks to the authors for addressing my questions. However, the authors seem not to have satisfactorily addressed the question raised in Point 2, namely, "authors should identify other markers that would help support that the mesenchyme were indeed specifically marked by Lsl1". Put in another way, what other markers does the mesenchyme express beside Lsl1? The authors have referred to several studies where Lsl1 was established as a mesenchyme marker. Do those studies also establish that Lsl1 is the only mesenchyme marker? If not, what makes the mesenchymes different from TRFs and CAFs?

The authors stated they have "used a total of five markers to identify fibroblasts", but that doesn't answer the question unless if their point is that mesenchymes are already tissue resident fibroblasts and CAFs and not necessarily give rise to those cells as the authors claim. Of course, if the mesenchymes give rise to fibroblasts, one can naturally expect certain molecular signatures/markers to be shared, but there should also be unique markers of the mesenchyme – maybe expressed more or less, but it is hard to argue that Lsl1 alone is the only markers of the mesenchyme. Perhaps an answer could come from or be supported by the single cell RNA seq data from which Fig. 2d was plotted. While it is interesting to identify cell types and subtypes and their origin, it is important that these plethora of cells are well defined to ensure reproducibility in independent studies.

No further comments besides the above.

Reviewer #3:

Remarks to the Author:

The authors have addressed my minor comments.

Reviewer #4:

Remarks to the Author:

The authors have made a great job in addressing all my concerns, and I have no more questions to bring up.

Regarding the definition of CAFs, I just think it is important to be stringent and not call cancer cells that has undergone EMT and started to express fibroblast markers for CAFs. CAFs are, per definition, not cancer cells and cannot originate from cancer cells. Normal epithelial cell can undergo EMT become CAFs. But if a tumor cell is undergoing EMT, it is a EMT cancer cell, not a CAF.

Reviewer #5:

Remarks to the Author:

I do not have any more comments. The authors have addressed all my queries.

“Many thanks to the authors for addressing my questions. However, the authors seem not to have satisfactorily addressed the question raised in Point 2, namely, “authors should identify other markers that would help support that the mesenchyme were indeed specifically marked by Lsl1”. Put in another way, what other markers does the mesenchyme express beside Lsl1? The authors have referred to several studies where Lsl1 was established as a mesenchyme marker. Do those studies also establish that Lsl1 is the only mesenchyme marker? If not, what makes the mesenchymes different from TRFs and CAFs?

The authors stated they have “used a total of five markers to identify fibroblasts”, but that doesn’t answer the question unless if their point is that mesenchymes are already tissue resident fibroblasts and CAFs and not necessarily give rise to those cells as the authors claim. Of course, if the mesenchymes give rise to fibroblasts, one can naturally expect certain molecular signatures/markers to be shared, but there should also be unique markers of the mesenchyme – maybe expressed more or less, but it is hard to argue that Lsl1 alone is the only markers of the mesenchyme. Perhaps an answer could come from or be supported by the single cell RNA seq data from which Fig. 2d was plotted. While it is interesting to identify cell types and subtypes and their origin, it is important that these plethora of cells are well defined to ensure reproducibility in independent studies.”

We greatly appreciate Reviewer #2’s further remarks, and we have included additional data and modified the text accordingly. Please see detailed response to each specific comment below.

“... “authors should identify other markers that would help support that the mesenchyme were indeed specifically marked by Lsl1”. Put in another way, what other markers does the mesenchyme express beside Lsl1?”

Besides *Isl1*, the splanchnic mesenchyme also expresses other well-established markers such as *Foxf1*, *Hand1* and *Pdgfra* (Han, *Nat Commun*, 2020; Rankin, *Cell Reports*, 2016). As suggested by the Reviewer, we performed an additional analysis using the published single cell RNA seq data from fetal foregut (Fig. 2). The result shows that *Isl1* and additional splanchnic markers *Foxf1*, *Hand1* and *Pdgfra* are enriched in the splanchnic cell cluster (**newly added Extended Data Fig. 2a**). To confirm this result at the protein level, we also included an additional immunostaining of FOXF1 using E9.5 tissue sections containing fetal foregut splanchnic mesenchyme (**newly added Extended Data Fig. 2b**).

“The authors have referred to several studies where *Lsl1* was established as a mesenchyme marker. Do those studies also establish that *Lsl1* is the only mesenchyme marker?”

We apologize for the confusion caused by the wording in the text. Importantly, it is the co-expression of *Isl1* with other known marker genes that distinguish the splanchnic mesenchyme from other cell compartments in the developing foregut. We added this clarification in line 120 “**ISL1, along with other established splanchnic markers**, was enriched in the splanchnic mesenchyme cluster”. We also changed the text in line 122 to “**ISL1 is one of the markers** of the splanchnic mesenchyme”.

“If not, what makes the mesenchymes different from TRFs and CAFs?” “The authors stated they have “used a total of five markers to identify fibroblasts”, but that doesn’t answer the question unless if their point is that mesenchymes are already tissue resident fibroblasts and CAFs and not necessarily give rise to those cells as the authors claim.”

The developmental timing of gene expression distinguishes the splanchnic mesenchyme from TRFs and CAFs (and other mesenchymal tissues). Defined temporal gene expression in

developing tissues is typically, but not always, transient as cells transit from progenitor states to well defined differentiation states, or transit through various physiological processes or diseases. For example, the splanchnic mesenchyme markers *Isl1*, *Hand1* and *Foxf1* are not robustly expressed in TRFs and CAFs, whereas *Pdgfra* was continuously expressed through development (**newly added Extended Data Fig. 7k**). To increase clarity, the text was modified in line 75: "...fibroblasts **in the adult pancreas** are defined as cells located in the stroma or interstitium, positive for markers such as VIMENTIN (VIM), PDGFR α , PDGFR β , PDPN and/or α SMA, ...".

This data is consistent with global transcriptome comparisons among the splanchnic mesenchyme, TRFs and CAFs. This analysis identified a group of genes highly expressed in the splanchnic mesenchyme but downregulated in TRFs, and another group of genes highly expressed in the splanchnic mesenchyme but downregulated in CAFs (**Fig 5e**). Taken together, we agree with the Reviewer that certain molecular signatures/markers are shared between fetal splanchnic mesenchyme and adult pancreatic fibroblasts, but there are also unique markers of the splanchnic mesenchyme that are expressed less in the adult fibroblasts (TRFs and/or CAFs).

Reviewers' Comments:

Reviewer #2:

Remarks to the Author:

Many thanks to the authors for addressing the concerns raised. I have no further comments.

“Reviewer #2 (Remarks to the Author): Many thanks to the authors for addressing the concerns raised. I have no further comments.”

Thank you.